# Morphological and molecular characterization of an *Elaeis oleifera* (H.B.K) Cortes germplasm collection located in Ucayali, Peru

**Alina Camacho-Villalobos**[1]*, **Fernando Serna**[2☯], **Jhofre Flores**[3☯], **Hector Flores**[2], **Paulo Manrique**[4], **Jorge Bendezu**[5]*

**1** Estación Experimental Agraria Pucallpa, Dirección de Desarrollo Tecnológico Agrario, Instituto Nacional de Innovación Agraria (INIA), Coronel Portillo, Ucayali, Perú, **2** Centro Experimental La Molina, Dirección de Recursos Genéticos y Biotecnología, Instituto Nacional de Innovación Agraria (INIA), Lima, Perú, **3** Facultad de Ciencias Agropecuarias, Universidad Nacional de Ucayali (UNU), Coronel Portillo, Ucayali, Perú, **4** Independent Researcher, **5** Estación Experimental Agraria Pucallpa, Dirección de Recursos Genéticos y Biotecnología, Instituto Nacional de Innovación Agraria (INIA), Coronel Portillo, Ucayali, Perú

☯ These authors contributed equally to this work.
* acamacho@inia.gob.pe (ACV); eduar156@yahoo.es (JB)

**Data Availability Statement:** All relevant data are uploaded to Figshare and accessible via the

## Abstract

The African oil palm (*Elaeis guineensis* Jacq) is a crop that is widely distributed in tropical regions around the world; however, this crop is subject to limitations such as rapid trunk growth and susceptibility to bud rot and red ring diseases particularly in South America. To overcome these limitations, national breeding and conservation programs have been established, and there is a need to identify parental palms from natural populations of the American oil palm (*E. oleifera* H.B.K. Cortes) with desirable yield and morphological traits (i.e., yield production and bunch number) and with high genetic diversity. However, in Peru the morphological and genetic data related to this important crop is limited. In this study, we characterized the morphological and yield and estimated the genetic diversity using 12 neutral microsatellite markers (simple sequence repeats, SSRs) across 72 oil palm individuals belonging to the *E. oleifera* germplasm collection located in the tropical region of Ucayali, Peru. Our results showed that morphological and yield traits explained approximately 40.39% of the variability within the Peruvian germplasm. Furthermore, Yield Production was highly correlated with two yield traits: Bunch Number (0.67) and Average weight per bunch (0.78). Based on the yield and morphological traits, a clustering analysis was performed and three phenotypic groups were identified (1, 2 and 3) in which groups 1 and 3 showed high scores associated primarily with yield traits. Microsatellite markers revealed 143 alleles, 11.92 ± 4.72 alleles per locus (A) and an expected heterozygosity (He) of 0.69 ± 0.045. A structural analysis identified three populations ($k = 3$), that were not related to the phenotypic groups. Interestingly, a multiple allele background was identified within the groups using multilocus and phylogenetic relationship analyses. This is the first Peruvian report regarding *E. oleifera* that shows preliminary data of the morphological and yield traits and genetic data, and highlight the importance of this information to set up future steps to national breeding strategies and improve the

following URL. dx.doi.org/10.6084/m9.figshare.
14132198.

**Funding:** This study was financially supported by Programa Nacional de Innovación Agraria (PNIA) of the Peruvian Government (http://www.pnia.gob.pe/) to ACV (contract 042_PI) and Programa de Jovenes Investigadores para el servicio de consultoría individual del I+D+i en el sector agrario to JB (INIA– PNIA-2020). Independent Research provided support for this study in the form of salary for PM. The specific roles of these authors are articulated in the 'author contributions' section. The funders had no role in study design, data collection and analysis, decision to publish, or preparation of the manuscript.

**Competing interests:** The authors have read the journal's policy and the authors of this manuscript have the following competing interests: PM is a paid employees of Independent Research. There are no patents, products in development or marketing products to declare. This does not alter our adherence to PLOS ONE policies on sharing data and materials.

conservation of genetic material of *E. oleifera*. Overall, these novel findings could contribute to the development of the local oil palm industry in Peru.

## Introduction

One of the most important characteristics of oil palms is the higher yield compared with other oil-producing crops worldwide [1–3]. In 2019, the oil palm production in Peru was 190,000 t for three Amazonian regions (San Martin, Ucayali, and Loreto) [4]. The African palm (*Elaeis guineensis* Jacq) is the most economically-important palm species and it is cultivated widely throughout the Peruvian Amazon forest [5–7], in contrast to the American palm (*E. oleifera* H.B.K. Cortes), which is native to the tropics of Central and South America and is well-recognized for its high yield and broader environmental adaptability [1, 2, 8].

The primary goals in the oil processing industry, particularly in the oil palm industry, are to increase the yield per hectare, improve oil quality and identify heterosis for traits such as disease resistance, fruit number, fruit weight, leaf length, and trunk traits [1, 2, 9]. For these reasons, some countries in South America have developed an *OxG* interspecific hybrid (*E. oleifera x E. guineensis*) program to produce individuals that have slow trunk growth and are tolerant to bud rot and red ring diseases using natural resources from their national germplasm collection [1, 2, 8–11].

Oil palm germplasm requires considerable financial support and a large land area, as one round of an oil palm conventional breeding cycle requires 10 years. Additionally, the development of new individuals requires 30–40 years [8]. Therefore, selected oil palms carrying genes for interesting morphological traits identified from the germplasm collection must be correctly preserved to ensure continuous access to and preservation of the genetic diversity with minimal redundancy among the individuals [12].

Molecular assays based on genomic DNA typification have been recognized as efficient tools to evaluate genetic diversity and develop genetic mapping studies supported by molecular markers. Among these molecular tools, amplification of the single sequence repeat (SSR) is popular due to advantages such as hypervariability, wide genomic distribution, co-dominant inheritance, a multi-allelic nature, and chromosome-specific locations; SSRs are also easier to implement than other DNA typification assays and are considered the most promising markers to understand the genetic structure of a population and identify potential parental genotypes in oil palms [1, 13–15].

The identification of interesting morphological traits and the knowledge of the genetic structure of the population as determinate by molecular markers in oil palm germplasm allows us to lead the effort to select potential parental genotypes to further breeding assays or tissue culture experiments to develop new individuals with improved traits; it also facilitates continued *ex-situ* conservation of the germplasm collection [16].

Unlike *E. guineensis*, *E. oleifera* can be harvested over a long period because of its slow trunk growth and its tolerance to pests and diseases in tropical regions; its oil also has a high level of unsaturated fatty acids. Therefore, *E. oleifera* germplasm is considered a strategic genetic resource because it encompasses a variety of desirable morphological traits, including those that confer adaptation to different tropical ecosystems such as the regions of South America [17]. The characterization of these genetic resources facilitates their effective utilization in national programs [18]. However, there is limited information regarding the phenotypic characterization and genetic diversity of *E. oleifera* in Peru [1, 7, 10].

In that context, the aim of this study was to characterize the *E. oleifera* (H.B.K) Cortes germplasm of a Peruvian collection using morphological traits and SSR markers to contributed with important preliminary data for setting up of further studies that will be related to the development of national breeding programs for the oil palm.

## Materials and methods

### Study area

This study was conducted at the Pacacocha annex of the Estación Experimental Agraria Pucallpa, Yarinacoha district, Coronel Portillo Provence, Ucayali region, Peru. Its geographical coordinates are 8˚21´9.75" S and 74˚33´5.59" W.

### Plant material

An American palm plantation *Elaeis oleifera* H.B.K. Cortes consisting of 72 four-year-old (from transplanting) plants in which each individual represents one different accession of this species were analyzed. The plant material belongs to the *Elaeis oleifera* germplasm maintained by Estación Experimental Agraria Pucallpa, Instituto Nacional de Innovación Agraria (INIA), Peru. All material was collected following Peruvian regulations [5, 19].

### Phenotyping

All experimental data obtained were related to the morphological and yield characteristics of an American palm (*Elaeis oleifera* H.B.K. Cortes) plantation that was laid out using a traditional pattern, a quincunx, with plant and row spacing of 9 x 9 m. Ten morphological characters were quantified: Trunk Diameter (TD was taken from the trunk circumference at the midsection from 10 cm below the crown height), Trunk Height (HT was estimated from the distance between the lowest green leaves and the fruit), Cup Coverage (CC, The leaves were counted one by one following the phyllotaxis of the trunk), Leaf Length ((LL) the leaf located in the middle third was taken, similar to the position of leaf number 17 in *E. guineensis*, once it was cut, the total length of petiole and length of the rachis were added to estimate the Leaf Length)), Petiole Length (PL was measured from petiole separated of the rachis in leaf number 17), Leaf Dry Weight (The dry weight was determined by chopping the leaf number 17 finely, and drying to constant temperature at 100-105˚C, then, the dry weight was multiplied by the number of leaves produced to estimate LDW), Foliar Area (FA represents the mean area per leaf multiplied by the number of leaves per palm this measure was obtained by measuring two leaves per palm), Leaflet length (LL was determinates in the largest leaflet in the leaf number 17 by measuring from apex to the insertion base of the rachis leaf), Leaflet diameter (LD was estimated by bending the largest leaflet in the middle and then width measure was taken) and Leaflet per Leaf (LXL, the number of leaflet per leaf was estimated by counting the leaflets in the leaf number 17 including rudimentary leaflets). The five yield characters were: Average Bunch Weight (ABW was determined by weighing the bunch at harvest time), Bunch Number (BN was represented by the number of bunches per palm collected during the first harvest), the 10-weight fruits (FW was estimated by weighing 10 fruits at random), fruit diameter (FD) and Yield per Palm (YP, this indicator was calculated to form the total production in kg of fruit per palm per year). Each measure was made using previously published methodologies [2, 20, 21]. And, the number of replications for each morphological trait was done every two weeks during the fourteen months of evaluation except for the LL, PL, LDW, and LD traits that were done once due to the implication of the use of destructive steps. The complete collected phenotyping data is presented in S1 File.

## Analysis of phenotypic data

Descriptive statistics were done calculated for morphological and yield traits (mean, standard deviation, coefficient of variation, maximum and minimum values). The phenotypic data were analyzed using Pearson's correlation coefficient (r) with $p \leq 0.05$ representing a significant correlation. A principal component analysis (PCA) was performed for all traits. A hierarchical cluster analysis was performed to identify phenotypic groups within the germplasm using the average linkage clustering method. All statistical procedures were performed using the R v3.42 software [22]. All quantitative data were visualized using GraphPad Prism 6.0 (GraphPad Software, La Jolla, CA, USA).

## DNA genotyping

The total DNA was extracted using the method of Doyle and Doyle [23] with modifications. Leaves were treated with liquid nitrogen, 1% (w/v) polyvinylpyrrolidone, and 3 mL CTAB extraction buffer. The DNA quality was verified with a 1% agarose gel using a GeneRuler 1 kb DNA molecular marker (Thermo Fisher Scientific, Carlsbad, CA, USA), and DNA samples were then quantified using UV-spectrophotometry in an Epoch 2 microplate spectrophotometer (Biotek, Winooski, VT, USA) following the manufacturer´s instructions.

Twelve microsatellite primers previously described [1, 13, 14, 24, 25] were employed in this study (Table 1). The polymerase chain reaction (PCR) was prepared with 5 ng of genomic DNA, 2 mM $MgCl_2$, 5 mM of a dNTP mix, 0.15 µM of each pair of primers (forward labeled with an M13 tag), and 1 unit of Kapa Taq DNA polymerase (KAPA Biosystems, Cape Town, WC, South Africa) in a final volume of 10 µL.

**Table 1. Description of 12 microsatellite markers to *E.oleifera* used in this study.**

| No | Microsatellite loci | Primer Sequences (5'-3') | Motif | Expected range size (bp) | Range reported in this study (bp) | Reported by |
|---|---|---|---|---|---|---|
| 1 | mEgCIR0353 | M13-ATTTCGTAAGGTGGGTGT CCTCCAAACTCCTCTGT | $(GT)_{11}(GA)_{15}$ | 80–102 | 81–118 | Ithnin *et al* 2017 |
| 2 | sM o00020 | M13-CCTTTCTCTCCCTCTCCTTTTG CCTCCCTCCCTCTCACCATA | $(AG)_{15}$ | 182–208 | 185–202 | Zaki *et al* 2012 |
| 3 | mEgCIR3282 | M13-GTAACAGCATCCACACTAAC GCAGGACAGGAGTAATGAGT | $(GA)_{20}$ | 232–272 | 234–252 | Ithnin *et al* 2017 |
| 4 | mEgCIR0067 | M13-TACACAACCCATGCACAT AAAAACATCCAGAAATAAAA | $(GA)_{17}$ | 135–187 | 139–179 | Billote *et al* 2001 |
| 5 | mEgCIR3886 | M13-TTCTAGGGTCTATCAAAGTCATAAG AGCCACCACCACCATCTACT | $(GA)_5GT$ $(GA)_{20}$ | 184–218 | 185–224 | Ithnin *et al* 2017 |
| 6 | mEgCIR0802 | M13-CTCCTTTGGCGTATCCTTTA TACGTGCAGTGGGTTCTTTC | $(GA)_{12}$ | 236–272 | 238–260 | Ithnin *et al* 2017 |
| 7 | mEgCIR0254 | M13-CCTTTTGTGCTTTCTTC GCTGTGCACTAGGTTTC | $(GA)_{18}$ | 148–179 | 153–179 | Arias *et al* 2012 |
| 8 | mEgCIR0437 | M13- CCAACCCAACCCAACATAAA GGTCCCGATCCCGTCTACT | $(CCG)_6$ | 196–206 | 103–293 | Arias *et al* 2012 |
| 9 | mEgCIR3285 | M13-AAGCAATATAGGTTCAGTTC TCATTTTCTAATTCCAAACAAG | $(GA)_{21}$ | 278–314 | 279–293 | Bakoumé *et al* 2007 |
| 10 | mEgCIR0018 | M13-CCTTATTTTCTTTGCTTACC TTCTATTTTATTTTCTTCCT | $(GA)_{18}$ | 158–177 | 141–198 | Billote *et al* 2001 |
| 11 | sM o00129 | M13-TTAGTATTGGGTGTGCATAAGTGG GCTTCCAGCTCCTCTTTCTACC | $(TTC)_{13}$ | 212–232 | 206–225 | Zaki *et al* 2010 |
| 12 | mEgCIR3546 | M13-GCCTATCCCCTGAACTATCT TGCACATACCAGCAACAGAG | $(GA)_{15}$ | 286–336 | 287–319 | Ithnin *et al* 2017 |

PCR was performed in a GeneAmp thermocycler (Thermo Fisher Scientific, Carlsbad, CA, USA). The cycling profile was as follows: an initial denaturation step for 5 min at 94˚C, followed by 30 cycles of denaturation at 94˚C for 1 min, annealing at 55˚C to 59˚C (touchdown) for 1 min, primer extension at 72˚C for 1 min and a final extension cycle of 5 min at 72˚C. The amplification products were labeled with fluorescent dyes (HEX or FAM) and assayed for the size on a 3130xl sequencer (Applied Biosystems, Foster City, CA, USA). The fragments were then scored with GeneMapper software v.3.7 (Applied Biosystems, Foster City, CA, USA) using default microsatellite settings in which bands smaller than 500 relative fluorescence units (rfu) were defined as background. The complete raw genetic data related to SSRs is showed in S1 File.

## Molecular data analysis

Microsatellite information was used to identify genetic diversity parameters with the Excel Microsatellite Toolkit [26]: expected heterozygosity (He), observed heterozygosity (Ho), the number of alleles per locus (A), inbreeding coefficient (Fis), polymorphic information content (PIC) and allelic richness (AR) were calculated for each marker in the total sample or the groups. To determine the population structure in the entire germplasm, an evaluation was performed using Structure software v.2.3.4. The parameters were set at a burn-in of 10,000 with 100,000 repeats for *K* values ranging from 1–10 [27]. Allelic multilocus genotypes were formed and Network analysis displaying the phylogenetic relationships between the phenotypic groups was performed with Network software v.10.0 (Fluxus Technology Ltd, Stanway, CT, England). Additionally, to test the effect of null alleles on the power of discrimination genotypes was addressed by the construction of a genotype accumulation curve with 1,000 iterations with the replacement of samples, while the effect on the He was addressed by a Monte Carlo test with 1,000 loci resampling considering the exclusion of loci with more than 10% of null alleles. All these procedures were tested using the R v.3.42 software.

## Results

### Phenotypic analysis

Descriptive statistics for the morphological and yield traits of the Peruvian *E. oleifera* germplasm (n = 72) are shown in Table 2. The morphological trait that exhibited the highest

**Table 2.  Descriptive statistics to morphological traits used in this study (n = 72).**

| Category | Trait | Abbreviation | Unit | Mean | SD | CV% | Minimum value | Maximun value |
|---|---|---|---|---|---|---|---|---|
| **Morphological** | Trunk Height | TH | cm | 514.5 | 59.93 | 11.65 | 290.0 | 662.0 |
| | Trunk Diameter | TD | cm | 206.8 | 31.32 | 15.15 | 148.0 | 309.0 |
| | Cup Coverage | CC | unit | 733.1 | 87.13 | 11.89 | 370.0 | 873.0 |
| | Leaf Length | LL | cm | 209.8 | 44.12 | 21.03 | 115.0 | 329.3 |
| | Petiole Length | PL | cm | 20.22 | 4.18 | 20.67 | 10.50 | 30.0 |
| | Leaflet per Leaf | LXL | unit | 95.79 | 19.05 | 19.89 | 68.0 | 200.0 |
| | Leaflet Length | LEL | cm | 60.76 | 8.41 | 13.84 | 42.80 | 90.29 |
| | Leaflet Diameter | LD | cm | 4.63 | 0.87 | 18.79 | 1.91 | 6.29 |
| | Leaf Dry Weight | LDW | kg | 3.82 | 0.83 | 21.73 | 1.49 | 5.31 |
| | Foliar Area | FA | cm$^2$ | 58,882.0 | 20,894.0 | 35.48 | 13,073.0 | 127,688.0 |
| **Yield** | Bunch Number | BN | unit | 15.65 | 5.32 | 33.99 | 1.0 | 31.0 |
| | Average Bunch Weight per oil palm | ABW | kg | 4.03 | 1.33 | 33.00 | 1.33 | 8.78 |
| | Yield per oil Palm per year | YP | mt | 9.21 | 4.32 | 46.91 | 0.60 | 22.60 |
| | Fruits 10-weight | FW | g | 103.0 | 22.76 | 22.10 | 54.90 | 155.0 |
| | Fruit Diameter | FD | cm | 1.06 | 0.14 | 13.21 | 0.69 | 1.33 |

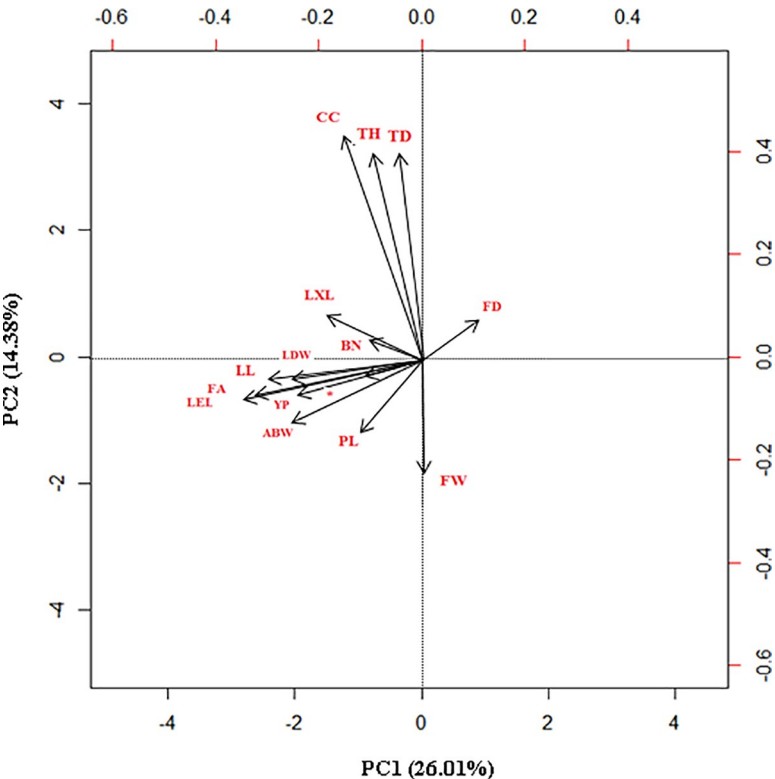

**Fig 1. Principal component analysis.** Principal components loading plot for the population of 72 *E.oleifera* for PC1 (26.01%) and PC2 (14.38%) calculated across 15 morphological and yield traits. * LD.

standard deviation (SD) was FA (SD: 20,894; min:13,073.0 –max: 127,688.0 cm$^2$; cv: 35.48%) and, one (YP) of the five yield traits showed high standard deviation (SD: 22.76; min: 0.6 – max: 22.60 mt; cv: 46.91%). These data show that both morphological and yield traits presented variability in the Peruvian germplasm.

A principal component analysis (PCA) was performed to identify the main traits that contributed to the dispersion of the values (Fig 1). Of the total variation in the Peruvian *E.oleifera* germplasm, was explained by two components: PC1 (26.01%), which was associated with five morphological traits (FA, LEL, LL, LXL, and LDW) and two yield traits (YP and ABW) and PC2, which was primarily associated with three morphological traits (TH, TD, and CC) and one yield trait (FW) explained the 14.38% of the variance in the Peruvian germplasm. The PCA analysis demonstrated that both the morphological and yield traits explained the variability in the oil palm individuals in this study (S1 Table in S1 File).

The Pearson correlation test was performed to identify the dependence among the analyzed traits (Fig 2). A significant correlation was detected between CC and both TD (0.53) and TH (0.73), and distinct correlations between FA, LL, LEL, and LDW (0.8–0.5) were identified; furthermore, YP was highly correlated with ABW and BN (0.78 and 0.67, respectively). Interestingly, only one weaker negative correlation (-0.4) was identified between a yield trait (FD) and a morphological trait (LXL), whereas other lower positive correlations (0.44–0.5) were identified between LEL and two yield traits (YP and ABW). Morphological trait values did not explain the yield traits values in the Peruvian *E.oleifera* germplasm, however, this study points out the potential relations (negative and positive) between YP, ABW, and FD and Leaflet traits (LXL and LEL).

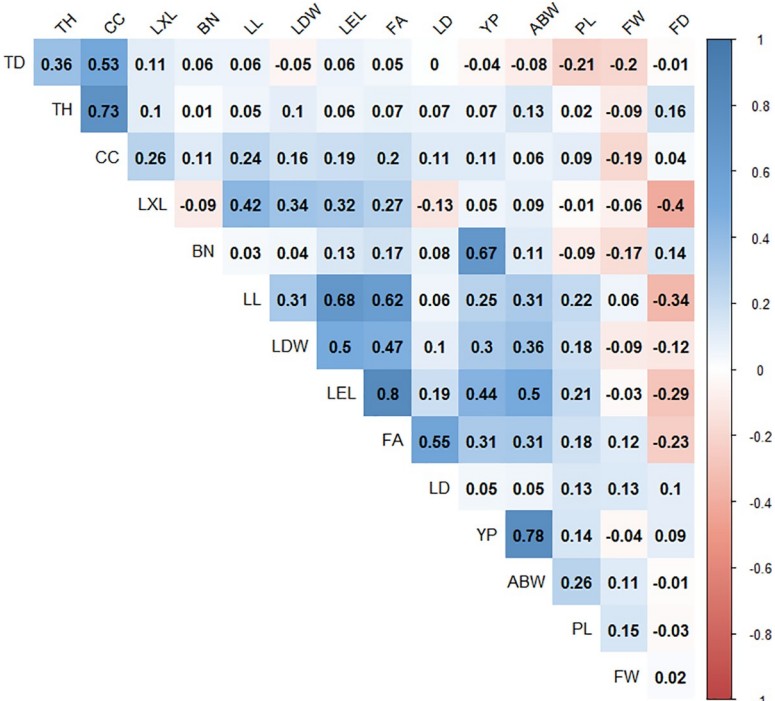

**Fig 2. Pearson correlations among 15 morphological traits in 72 individuals from Peruvian *E.oleifera* germplasm.**
The acronyms in the figure for each trait are described in the materials and methods section. Colored boxes indicate
significant correlations (p<0.05). Blue boxes indicate a positive correlation; Red boxes indicate a negative correlation
and white boxes indicate no correlation. The colored vertical bars indicate the level of correlation (-1 to 1).

To evaluate the level of phenotypic similarity, a hierarchical cluster analysis was performed
on the 72 oil palm individuals from this Peruvian germplasm. Overall, we identified three
main groups or clusters (1, 2, and 3) based on their morphological and yield traits. Groups 1
and 2 were more closely related to each other than to group 3; however, twelve oil palms were
positioned separately or were identified as groups with one or two individuals (Fig 3). The

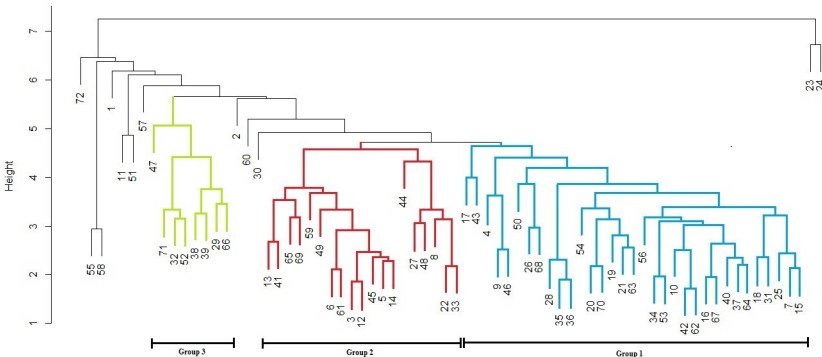

**Fig 3. Hierarchical cluster analysis of the *Elaeis oleifera* Peruvian germplasm collection calculated across 15
morphological and yield traits.** Seventy-two individuals were clustered using the average linkage clustering method.
Phenotypic groups were identified and their branches are colored blue (Group 1, n = 33), red (Group 2, n = 19) and
green (Group 3, n = 8). Some individuals were separated out in the comparative analysis
(1,2,11,23,24,30,51,55,57,58,60, and 72).

three groups showed significant differences in most of the traits except to TH, PL, and FD (Fig 4). Group 3 showed the highest mean values of the traits related to the leaf (LL, LXL, LEL, LDW, and FA), and both groups 1 and 3 showed high values related to three yield traits (BN, ABW, and YP). Additionally, group 2 exhibited the lowest values of LL, LDW, LEL, and FA (Fig 4).

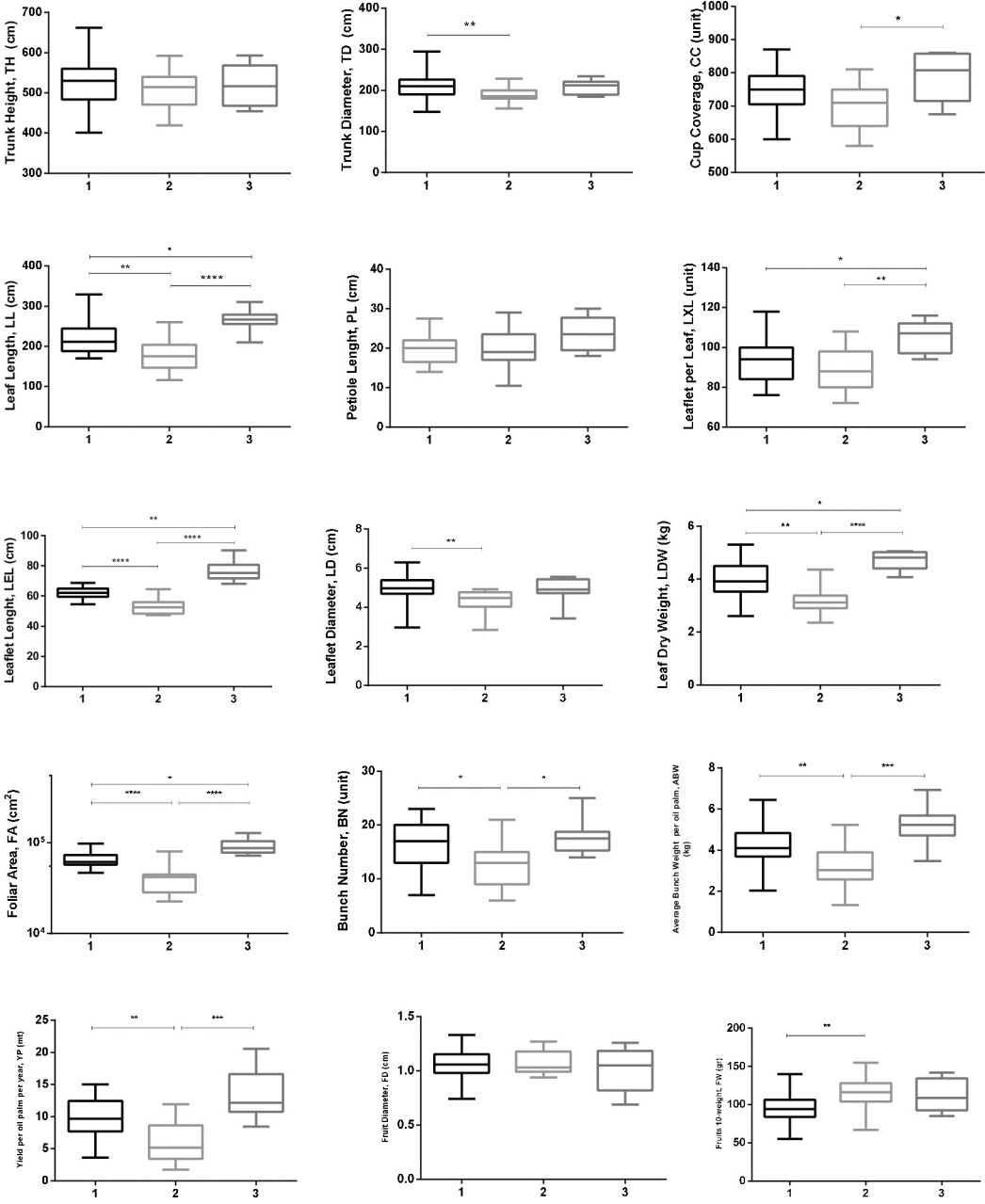

**Fig 4. Box-plot graphs of the morphological (TH, TD, CC, LL, PL, LXL, LEL, LD, LDW, and FA) and yield traits (BN, ABW, FD, YP and FW) for the Phenotypic groups previously identified (1, (n = 33); 2, (n = 19); and 3, (n = 8)).** A non-parametric test (*Kruskal-Wallis*) was used, multiple comparisons were run and corrected using Dunn's test. Significance levels are indicated by horizontal bars. **** < 0.001; ** <0.01 and * < 0.05.

## Molecular analysis

The molecular characterization of the Peruvian *E.oleifera* germplasm was conducted using 12 neutral microsatellites (Table 1). The molecular markers generated 143 scorable alleles across 72 *E. oleifera* (H.B.K.) Cortes palms. The genetic diversity was calculated for the entire germplasm and for the three groups separately (Table 3). Overall, the number of alleles ranged from 4 (mEgCIR3546) to 20 (mEgCIR0018), the mean number of scored alleles per locus (A) was $11.92 \pm 4.72$, the mean polymorphism information content (PIC) was 0.65, the expected heterozygosity (He) was $0.69 \pm 0.045$, the genetic variation estimation (Ho) was $0.37 \pm 0.02$ and consequently, the Fis showed a positive value (0.46).

Using the same method as for the complete germplasm, the genetic diversity was determined for each group (1, 2, and 3); group 3 exhibited the lowest number of alleles ($4.17 \pm 1.85$) compared with the other groups ($8.83 \pm 3.33$ and $6.50 \pm 3.78$). For the other genetic parameters (He, Ho, and Fis), the values obtained were similar between the total germplasm and the identified groups ($p > 0.05$).

The frequency of null alleles was estimated between 0.01 and 0.48 for nine microsatellites (S2 Table in S1 File). Then, five microsatellites that presented null alleles were excluded to perform a genotype accumulation curve and expected heterozygosity tests to identify the effect of null alleles in the subsequent analysis. The analysis showed that the use of at least seven microsatellites could identify the totality of the 72 multilocus genotypes identified (S1 Fig in S1 File), also the inclusion or exclusion of the null alleles in the analysis did not alter the final He ($p < 0.05$) in the Peruvian germplasm (S2 Fig and S3 Table in S1 File).

To infer the population structure, an analysis was performed in Structure using all the oil palms in this study. The software predicted three and four subpopulations ($k = 3$ and 4) as the best models for the Peruvian germplasm (Fig 5), which is consistent with the high Fis value indicating the presence of population structure; however, these subpopulations were not related to the phenotypic groups previously identified by hierarchical clustering (Fig 3).

The microsatellite marker data established an allelic multilocus for each oil palm to assess the gene flow across the Peruvian *E.oleifera* germplasm. The Network analysis and gene flow display the phylogenetic relationship among the oil palms of this study colored by phenotypic-group origin (Fig 6). The allelic multilocus genotypes for all palms were distributed across the network and indicated high diversity within the three phenotypic groups. Four potential sub-groups that shared similar molecular markers were also identified; the individuals from group 3 did not exhibit sub-groups in the network due to the low quantity of the samples in that group ($n = 8$).

## Discussion

### Morphological and yield characterization

The national breeding program for oil palm must achieve two important goals; first, to incorporate the traits of interest into the next generation of oil palms for future access and second, to retain the genetic variability for selection gains in the future [8]. To contribute to these goals, our study addressed to the morphological and molecular characterization of an *Elaeis oleifera* (H.B.K) Cortes germplasm collection located in Ucayali, Peru.

Two of the morphological traits were related to the trunk dimensions (height—TH) and (diameter—TD) to identify oil palms with a smaller stature that permits easy access to the fruits and increases its commercial life because slow growth saves resources at the expense of the vegetative growth [2, 9]. In this study, the mean TH was $514 \pm 59.93$ cm, which is similar

**Table 3. Genetic statistics for the germplasm and the phenotypic groups using 12 microsatellites (SSR).**

| Population/ Locus name | N° Alleles scored | He | PIC | Allelic Richness (AR) | N° Alleles (A) ± sd | He ± sd | Ho ± sd | Fis |
|---|---|---|---|---|---|---|---|---|
| **Germplasm (n = 72)** | | | | | | | | |
| mEgCIR0353 | 11 | 0.72 | 0.68 | 10.02 | 11.92 ± 4.72 | 0.69 ± 0.045 | 0.37 ± 0.02 | 0.46 |
| sM o00020 | 15 | 0.67 | 0.64 | 11.46 | | | | |
| mEgCIR3282 | 10 | 0.72 | 0.68 | 9.62 | | | | |
| mEgCIR0067 | 9 | 0.63 | 0.58 | 8.43 | | | | |
| mEgCIR3886 | 7 | 0.54 | 0.45 | 6.03 | | | | |
| mEgCIR0802 | 9 | 0.66 | 0.63 | 9.00 | | | | |
| mEgCIR0254 | 19 | 0.90 | 0.88 | 17.36 | | | | |
| mEgCIR0437 | 15 | 0.80 | 0.77 | 11.93 | | | | |
| mEgCIR3785 | 13 | 0.74 | 0.72 | 12.10 | | | | |
| mEgCIR0018 | 20 | 0.86 | 0.84 | 15.60 | | | | |
| sM o00129 | 11 | 0.72 | 0.68 | 9.73 | | | | |
| mEgCIR3546 | 4 | 0.31 | 0.28 | 3.71 | | | | |
| **Total** | 143 | | *0.65 | | | | | |
| **Group 1 (n = 33)** | | | | | | | | |
| mEgCIR0353 | 10 | 0.69 | 0.64 | 1.64 | 8.83 ± 3.33 | 0.69 ± 0.03 | 0.38 ±0.03 | 0.44 |
| sM o00020 | 11 | 0.66 | 0.62 | 1.66 | | | | |
| mEgCIR3282 | 8 | 0.77 | 0.72 | 1.77 | | | | |
| mEgCIR0067 | 5 | 0.57 | 0.51 | 1.57 | | | | |
| mEgCIR3886 | 5 | 0.49 | 0.43 | 1.49 | | | | |
| mEgCIR0802 | 8 | 0.71 | 0.65 | 1.71 | | | | |
| mEgCIR0254 | 13 | 0.91 | 0.88 | 1.91 | | | | |
| mEgCIR0437 | 11 | 0.79 | 0.74 | 1.79 | | | | |
| mEgCIR3785 | 11 | 0.77 | 0.73 | 1.77 | | | | |
| mEgCIR0018 | 14 | 0.86 | 0.83 | 1.86 | | | | |
| sM o00129 | 6 | 0.72 | 0.66 | 1.72 | | | | |
| mEgCIR3546 | 4 | 0.28 | 0.26 | 1.28 | | | | |
| **Total** | 106 | | | | | | | |
| **Group 2 (n = 19)** | | | | | | | | |
| mEgCIR0353 | 8 | 0.75 | 0.71 | 1.75 | 6.50 ± 3.78 | 0.66 ± 0.05 | 0.36 ± 0.03 | 0.45 |
| sM o00020 | 8 | 0.69 | 0.64 | 1.69 | | | | |
| mEgCIR3282 | 5 | 0.65 | 0.58 | 1.65 | | | | |
| mEgCIR0067 | 2 | 0.46 | 0.35 | 1.46 | | | | |
| mEgCIR3886 | 2 | 0.58 | 0.37 | 1.51 | | | | |
| mEgCIR0802 | 4 | 0.57 | 0.50 | 1.57 | | | | |
| mEgCIR0254 | 13 | 0.90 | 0.86 | 1.90 | | | | |
| mEgCIR0437 | 8 | 0.80 | 0.74 | 1.79 | | | | |
| mEgCIR3785 | 9 | 0.68 | 0.65 | 1.68 | | | | |
| mEgCIR0018 | 12 | 0.88 | 0.84 | 1.87 | | | | |
| sM o00129 | 5 | 0.69 | 0.61 | 1.68 | | | | |
| mEgCIR3546 | 2 | 0.34 | 0.28 | 1.34 | | | | |
| **Total** | 78 | | | | | | | |
| **Group 3 (n = 8)** | | | | | | | | |
| mEgCIR0353 | 4 | 0,68 | 0.57 | 1.70 | 4.17 ± 1.85 | 0.70 ± 0.07 | 0.39 ± 0.06 | 0.44 |
| sM o00020 | 4 | 0,51 | 0.44 | 1.65 | | | | |
| mEgCIR3282 | 4 | 0,80 | 0.67 | 1.73 | | | | |
| mEgCIR0067 | 4 | 0,86 | 0.70 | 1.58 | | | | |

(*Continued*)

**Table 3.** (Continued)

| Population/ Locus name | N° Alleles scored | He | PIC | Allelic Richness (AR) | N° Alleles (A) ± sd | He ± sd | Ho ± sd | Fis |
|---|---|---|---|---|---|---|---|---|
| mEgCIR3886 | 2 | 0,48 | 0.35 | 1.51 | | | | |
| mEgCIR0802 | 2 | 1,00 | 0.37 | 1.66 | | | | |
| mEgCIR0254 | 5 | 0,80 | 0.70 | 1.89 | | | | |
| mEgCIR0437 | 6 | 0,83 | 0.75 | 1.79 | | | | |
| mEgCIR3785 | 3 | 0,43 | 0.37 | 1.71 | | | | |
| mEgCIR0018 | 8 | 0,89 | 0.81 | 1.86 | | | | |
| sM o00129 | 6 | 0,83 | 0.75 | 1.73 | | | | |
| mEgCIR3546 | 2 | 0,23 | 0.20 | 1.29 | | | | |
| **Total** | 50 | | | | | | | |

He: Expected heterozygosity; Ho: observed heterozygosity, A: alleles per locus, Fis: Inbreeding coefficient, PIC: Polymorphic information content. sd: standard deviation. n: number of accessions.

*Total PIC was calculated.

to the results of a previous study conducted in Huánuco, Peru [7]; however, this value is higher than that of other South American regions [2, 17, 28, 29].

There is a relation between trunk diameter and the consumption of dry-vegetative material and a low rate of production [9], and the mean value obtained for the *E. oleifera* germplasm was higher compared with that of other studies inside [7] and outside Peru [17, 28, 29].

The primary phenotypic differences within this Peruvian germplasm were based on three morphological traits, trunk height (TH), trunk diameter (TD) and cup coverage (CC), and one yield trait (FW, Fruits 10-weight) which explained 40.4% of the total variation, in contrast, the

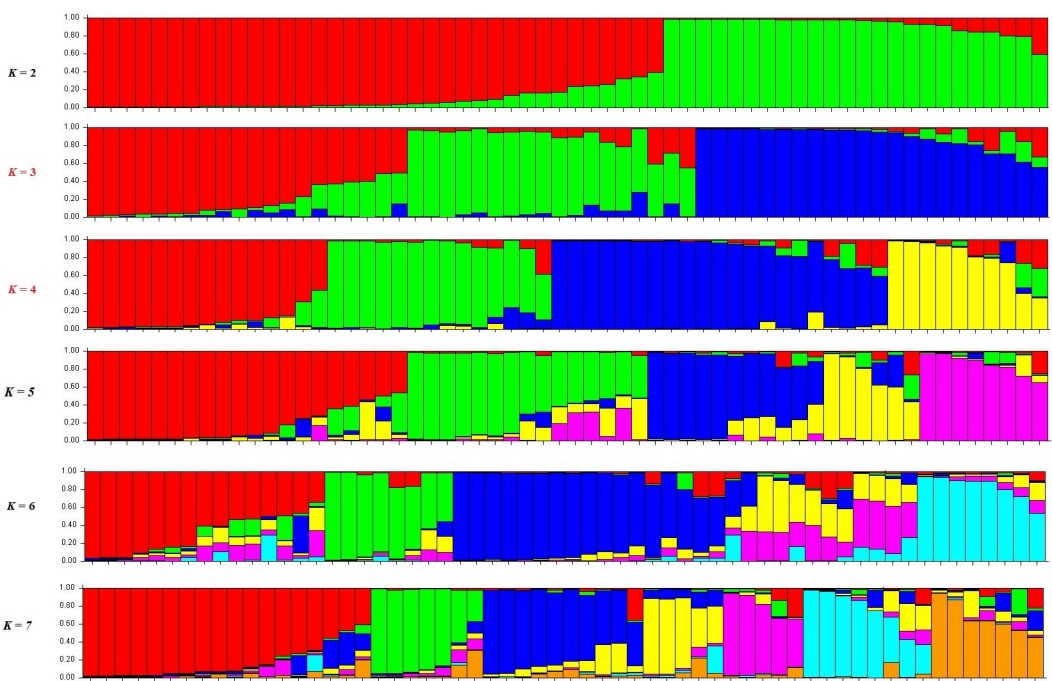

**Fig 5. Genetic clustering analysis using structure.** The graph depicts the clustering models when the individuals were assigned to 2 to 7 clusters (k = 3 and 4 were the best models and are labeled in red).

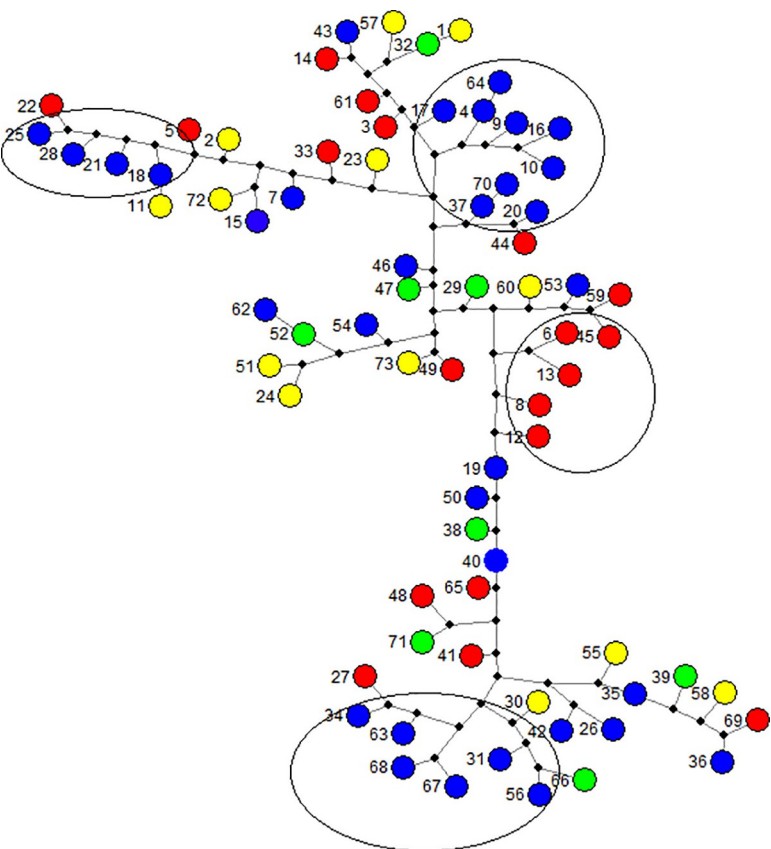

**Fig 6. Network analysis displaying the phylogenetic relationships within the *E. oleifera* germplasm arranged by the hierarchical cluster analysis as previously described in this study.** The individuals are represented by ID numbers and the color represents the phenotypic group (Group: 1, blue; 2, red; and 3, green). Yellow circles indicate individuals with no relation to the phenotypic groups (genotypes excluded). The black nodes denote the presence of one virtual genotype. Dashed circles indicate subgroups.

variation in an *E. oleifera* Colombian germplasm was explained by four morphological traits (trunk height, leaf area, foliar area, and leaf dry weight) [2] however, the age of both plantations could explain this differences.

To identify interesting traits in the Peruvian germplasm, comparisons were made between all morphological and yield traits, and some interesting associations were identified. Two yield traits (ABW and BN) showed a high correlation with YP, which is consistent with the results of a Colombian report [1, 2, 8], which identified relationships between yield and morphological traits.

Each leaf in an oil palm has an inflorescence whose size and development depend on the physiological state of the individual oil palm, hence the importance of leaf attributes to bunch yield [30]. In our study of the Peruvian germplasm, there was a weak positive correlation between one trait related to leaflet length and yield production (0.44) and to the bunch weight (0.5), which corroborates the importance of the leaf in production.

A larger foliar area in oil palm is related to high photosynthetic activity, which could be related to higher production [2], and the leaflet number contributes directly to this trait [17]. However, in this study, we observed a negative correlation (-0.4) between LXL and fruit diameter; further investigation is necessary to clarify this finding.

One limitation of the present study was the lack of characterization of the fruit; this would determine the correlation of increased mesocarp, kernel oil yields, and oil quality with the number and weight of bunches to enhance the future core germplasm [2, 3, 8, 16].

In this study, yield production (YP) was used to identify important phenotypic traits; however, YP did not contribute extensively to the total variability of the germplasm in comparison with other morphological traits (TH, TD, and CC) and another yield trait (FW); for this reason, we conducted a hierarchical clustering analysis to identify phenotypic groups with different traits.

The clustering analysis separated our germplasm into three groups (1, 2, and 3) that were identified by significant trait differences among them. Interestingly, group 3 exhibited high mean values of 5 traits related to the leaf (LL, LXL, LEL, LDW, and FA); this "yield phenotype" group could have a higher photosynthetic capacity and greater nutrient absorption by the roots [28, 31].

Similarly, groups 1 and 3 exhibited a phenotype different from group 2 in regards to yield traits such as BN, ABW, and YP. According to previous reports [2, 8, 9] and the results in this study, yield production is related to the average weight per bunch (ABW), and the bunch number (BN); this phenotype could be explained by the high values of morphological traits in groups 1 and 3 compared with those of group 2.

Additionally, there was a difference between the phenotypic groups (1 and 2, p<0.01) in trunk diameter (TD), excluding the age factor, this is an interesting trait to select within the Peruvian germplasm, in contrast to reports by others studies [17, 20, 21].

## Genetic characterization

Genetic characterization was conducted for the Peruvian germplasm to identify alleles, population structure, and genetic relationships across the 72 *E. oleifera* individuals. Previous studies on the genetic diversity of *E. oleifera* using distinct molecular markers are limited in South America [8, 10, 15, 32] and there is very limited information regarding *E. oleifera* Peruvian germplasm located in Loreto region, Peru [1].

In this study, we employed neutral twelve microsatellite markers (SSRs) to study germplasm located in Ucayali region (located 634 km from Loreto); a total of 143 alleles were scored, which is high compared with others studies using Brazilian and Peruvian germplasm [1]. Two microsatellites (mEgCIR0018 and mEgCIR0254) showed a high PIC in our samples and both markers were applied to an oil African palm (*E. guinenesis* (H.B.K) Cortes) germplasm that exhibited moderate PIC values [33]. This difference compared with the other ten microsatellites in our study could be explained due to the variation within the Arecaceae family genomes that primarily reflect changes in the number of repeat motifs in the SSR region combined with indels and base substitutions [15]. Interestingly, the mEgCIR3546 marker with the lowest PIC in our study was reported to have the same value in another study using samples from Loreto, Peru [13]. However, mEgCIR0018, mEgCIR0254 and mEgCIR3546 exhibited a bi-nucleotide motif ($(GA)_{15}$ and $(GA)_{18}$) in contrast to that previously described [15] in which a high PIC value related to a tri-nucleotide motif was reported.

Null alleles were previously reported in Peruvian samples with frequencies around 0.06 and 0.21 [1], and in this study were reported between 0.01 to 0.48, however, the presence of five microsatellites with more than 10% of null alleles (or missing information) do not alter the estimation of the genetic diversity, in terms of He, of the germplasm of 72 samples (S3 Table in S1 File). These null alleles are produced by putative mutations that occur at the microsatellite binding sites that could prevent primer annealing and in consequence, the lack of amplification of the PCR products [8] therefore further primer development could be necessary for the local or regional *E. oleifera* germplasm.

The expected heterozygosity (He) indicates the level of genetic diversity in a population, and it is related to the number of alleles; the calculated value for the germplasm in Ucayali, Peru was 0.69 ± 0.045. This value is the highest observed compared with other tropical regions in Central and South America (Ecuador, Panama, Costa Rica, Colombia, Brazil, and Loreto, Peru) [1, 8] and this information will be important to the development of the future studies related to set up core collection in Peru.

However, the Ho calculated in our study (0.37 ± 0.02) was low compared with estimated He. The Ho value reported in this study is similar to other reports [1], the calculated Fis was consistent with the low level of diversity observed (Ho) and could be explained by the advanced breeding population, which had undergone several cycles of selfing [15]. This selfing event was related to a low of natural pollination in *E. oleifera*, because the peduncular bracts that cover the bunch restrict the work of pollinating insects [34].

The Structure analysis indicated three or four populations ($k$ = 3,4) across the 72 oil palms, and previous studies have suggested that population size of approximately 20–30 individuals will accurately represent the genetic diversity present in oil palms [35]. Using three populations ($k$ = 3), there was no correlation with the phenotypic groups previously identified, because each group showed a multiple allele background that contributed to the final phenotype. An allelic pool will be an important genetic gain through conventional breeding [8] and these findings corroborate the high Fis value identified in this study, which could be explained by (i) high levels of inbreeding [3] and (ii) the limited number of individuals who compose the population versus the number of alleles obtained [36].

The importance of the core collection derived from germplasm is the preservation of genetic in a small number of individuals. The implementation of such a collection is an excellent option for oil palm programs because it allows more effective access to the genetic material [8, 36]. To establish the future core it is important to achieve important data such as the number of alleles (A), the genetic diversity (He) using neutral molecular markers, and the specific morphological and yield traits (1). In this study, three phenotypic groups (1, 2 and 3) were identified in which exhibited similar He values (group: 1 = 0.69 ± 0.03; group 2 = 0.66 ± 0.05 and group 3 = 0.70 ± 0.07; p<0.05) and the number of alleles (A) across the three groups were also similar (group 1 = 8.83 ± 3.33; group 2 = 6.50 ± 3.78 and group 3 = 4.17 ± 1.85; p<0.05). These results will help to identify ideal populations that show the allelic richness and exhibit high genetic variation [8], however, a population of around 20–30 individuals per each identified group should be considered to further evaluations employing the same set of neutral molecular markers presented in this study to avoid misinterpretation in the genetic diversity parameters.

The analysis of the phylogenetic relationships across the Peruvian germplasm showed a multiple allelic background for each phenotypic group (Fig 6). Our results show the need to conserve those alleles as part of the conservation of genetic diversity [1]. Additionally, subgroups were identified that were composed of individuals who shared similar alleles (multilocus), this information is important because those subgroups should be considered in further assays optimizing interpopulation heterosis [32]. More studies using a large set of microsatellites (≥ 20 SSRs) or single nucleotide polymorphisms (SNPs) are needed to identify genomic regions with genes that are related to important morphological traits and traits that are related to yield in oil palm [2].

The use of selected molecular markers could contribute to the modernization of plant breeding programs because such tools facilitate the selection of promising accessions at early stages (i.e., greenhouse conditions) and therefore circumvent the long breeding cycle of the oil palm [2, 16]. Moreover, the limited economical resources in many laboratories in South America could interrupt the adoption of these molecular tools. Recently, there are studies in *E*.

| Groups | Morphological and yield traits | | E.oleifera selected accessions | | | Genetic diversity parameters (± SD) |
|---|---|---|---|---|---|---|
| **1 (n=33)**<br><br>17,43,4,9,46, 50,26,68,28, 35,36,54,20, 70,19,21,63, 56,34,53,10, 42,62,16,67, 40,37,64,18, 31,25,7,15 | ↕LL ↕LEL ↕LDW ↕FA | ↑BN ↑ABW ↑YP |  |  |  | A: 8.83 ± 3.33<br>He: 0.69 ± 0.03 |
| **2 (n= 19)**<br><br>13,41,65,69, 59,49,6,61, 3,12,44,45, 5,14,27,48, 8,22,33 | ↓LL ↓LEL ↓LDW ↓FA | |  |  |  | A: 6.50 ± 3.78<br>He: 0.66 ± 0.05 |
| **3 (n=8)**<br><br>71,32,52,38, 39,29,66,47 | ↑LL ↑LXL ↑LEL ↑LDW ↑FA | ↑BN ↑ABW ↑YP |  |  |  | A: 4.17 ± 1.85<br>He: 0.70 ± 0.07 |

**Fig 7. The morphological, yield, and genotypic differences between the three groups (1,2 and 3) formed within *E. oleifera* Peruvian germplasm.** A: Allele numbers per locus; He: expected heterozygosity; SD: standard deviation. (↕median value; ↓Low value; ↑High value were assigned according to Fig 4). Number underlined indicates the selected accessions in the pictures.

*oleifera* that show that useful information is achieved when is employed a limited set of microsatellites (< 20 SSRs) [1, 8] and according to our analysis could be necessary at least 7 microsatellites in the Peruvian germplasm to carry up the complete analysis of genetic diversity (S1 Fig and S3 Table in S1 File), however, many studies are necessary to corroborate these findings.

Understanding the relationship between morphological traits and genetic diversity is essential for the conservation of the *E. oleifera* germplasm [1]. Our results identified three phenotypic groups with different traits; two (1, n = 33 and 3, n = 8 designated "yield groups") of these groups exhibited interesting yield traits (bunch number and average weight per bunch) and optimal traits related to leaf morphology. Additionally, the yield in both groups was represented by a similar quantity of alleles (A) and high genetic diversity (He) (Fig 7).

This study described the morphological and genetic characterization of one Peruvian *E. oleifera* germplasm; these data are a starting point for further steps related to the measurement of photosynthetic activity [17] and the physicochemical characterization of bunches [1, 11] and their relation to the phenotypic groups identified in this study. However, this information should be taken carefully due to the age of the germplasm (four-year-old) and further studies are necessary to complement it, by including new data trait (morphological and yield) corresponding to the next four years in according to other previous studies [1, 20]

Finally, this novel information presented here demonstrates that the 72 accessions of the Peruvian *E. oleifera* located in Ucayali region of Peru represent a useful preliminary genetic resource to promote the use of the genetic material and design further investigations to establish national breeding strategies and conservation programs for this important crop in the tropical regions of Peru and South America.

## Supporting information

**S1 Database. Morphological—yield traits and SSRs_Camacho-Villalobos A et al 2020.** (XLSX)

**S1 File.** (DOC)

## Acknowledgments

We thank MSc. Glendy Sanchez for her critical and constructive recommendations to the initial reports of this study. Also, we thank Bio Transfer SAC (Peru) to support us in the English editing service. The funders had no role in the study design, data collection, and analysis, decision to publish, or preparation of the manuscript.

## Author Contributions

**Conceptualization:** Alina Camacho-Villalobos.

**Data curation:** Alina Camacho-Villalobos, Jhofre Flores, Jorge Bendezu.

**Formal analysis:** Alina Camacho-Villalobos, Fernando Serna, Jorge Bendezu.

**Funding acquisition:** Alina Camacho-Villalobos.

**Investigation:** Alina Camacho-Villalobos.

**Methodology:** Alina Camacho-Villalobos, Fernando Serna, Jhofre Flores, Hector Flores.

**Project administration:** Alina Camacho-Villalobos.

**Resources:** Alina Camacho-Villalobos.

**Software:** Paulo Manrique.

**Supervision:** Alina Camacho-Villalobos.

**Validation:** Alina Camacho-Villalobos, Fernando Serna, Jorge Bendezu.

**Visualization:** Alina Camacho-Villalobos, Fernando Serna, Paulo Manrique, Jorge Bendezu.

**Writing – original draft:** Alina Camacho-Villalobos, Jorge Bendezu.

**Writing – review & editing:** Alina Camacho-Villalobos, Fernando Serna, Jhofre Flores, Hector Flores, Paulo Manrique, Jorge Bendezu.

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
