## [Decision Letter · Decision Letter 0]

7 Jul 2020

PONE-D-20-17719

Morphological and molecular characterization of an Elaeis oleifera (H.B.K) cortes germplasm collection located in Ucayali, Peru.

PLOS ONE

Dear Dr. Bendezu,

Thank you for submitting your manuscript to PLOS ONE. After careful consideration, we feel that it has merit but does not fully meet PLOS ONE’s publication criteria as it currently stands. Therefore, we invite you to submit a revised version of the manuscript that addresses the points raised during the review process.

We look forward to receiving your revised manuscript.

Kind regards,

Tzen-Yuh Chiang

Academic Editor

PLOS ONE

Journal Requirements:

'The authors have declared that no competing interests exist.'

We note that one or more of the authors are employed by a commercial company: Independent research.

Reviewers' comments:

Reviewer's Responses to Questions

**Comments to the Author**

1. Is the manuscript technically sound, and do the data support the conclusions?

Reviewer #1: Yes

Reviewer #2: No

2. Has the statistical analysis been performed appropriately and rigorously? 

Reviewer #1: Yes

Reviewer #2: Yes

3. Have the authors made all data underlying the findings in their manuscript fully available?

Reviewer #1: Yes

Reviewer #2: Yes

4. Is the manuscript presented in an intelligible fashion and written in standard English?

Reviewer #1: Yes

Reviewer #2: Yes

5. Review Comments to the Author

Reviewer #1: 1. The study focuses on morphological, yield traits and genetic diversity of E.oleifera. However, the study on fruit related traits, which is an important commercial trait in oil palm is missing. Furthermore, according to descriptive statistics (Table 2), morphological traits has reduced variability than yield traits whereas yield traits has reduced variability than morphological traits in PCA analysis. There was a conflict between these two analyses. Is there any explanation for the observed results?

2. The author described the importance of relationship between the morphological traits and genetic diversity. According to their results, groups 1 and 3 were more closely related in morphological and yield traits whereas groups 1 and 2 were more closely related in genetic diversity analysis (number of alleles). Would appreciate more explanation for these results.

3. The abstract can be made more reflective of the total paper. The authors need to highlight what are altogether the novel results of their study.

4. Although previously published methodologies were referred, the methods and the number of replicates for measurement of phenotypic data in materials and methods section should be briefly described.

5. In the results section (Page 8, Line #186-187), it is not clear whether yield traits are compared to morphological traits or not. Please re-write that part.

6. Page 11, Line # 261, the word "wich" should be "which".

7. Page 581, Table 4, the word "morphologycal" should be "morphological".

8. Page 262, Line 262-263, please clearly describe the sentences.

9. English is sometimes odd, please improve it.

Reviewer #2: The manuscript describes genetic diversity of Elaeis oleifera assembled from an area in Peru by means of 12 SSRs and selected morphological traits. The manuscript was written in readable English language but with minor spelling mistakes and some grammatical errors. These errors are however correctable.

I have specific concerns as below

1. The phenotypic data was collected at very young age, when the palms are 4 years old, as described by authors in method section. It should be cautioned that analysis and interpretation of such data can be erroneous and misleading. It is very important to note that For oil palm specifically, standard phenotypic data collection is carried out for at least 8 years, starting from palms at 2 or 3 years old. Measuring vegetative traits such as height, frond length is carried out once on palms at 8th year old. I am doubtful on the results presented especially if it is going to be used for the purposes of selection and breeding.

2. The challenge of using SSRs for genetic analysis is the occurrence of null alleles. Null alleles introduce error in assessment of genetic diversity, parentage analysis etc. The authors can carry out relevant analysis to identify null alleles, exclude them from dataset prior to genetic diversity analysis.

3. The number of SSRs applied is limited. The trend of publication nowadays applied hundreds or even thousands of markers for diversity analysis. I encourages authors to add more markers to achieve reasoanably acceptable statistics to support any conclusions made from the analysis.

6. PLOS authors have the option to publish the peer review history of their article (what does this mean?). If published, this will include your full peer review and any attached files.

Reviewer #1: No

Reviewer #2: No

---

## [Author Response · Author response to Decision Letter 0]

19 Aug 2020

Journal Requirements:

1. Comment: When submitting your revision, we need you to address these additional requirements. Please ensure that your manuscript meets PLOS ONE's style requirements, including those for file naming. The PLOS ONE style templates can be found at https://journals.plos.org/plosone/s/file?id=wjVg/PLOSOne_formatting_sample_main_body.pdf and https://journals.plos.org/plosone/s/file?id=ba62/PLOSOne_formatting_sample_title_authors_affiliations.pdf

Response and change: We agree with the comment. We have revised diligently our manuscript and adapted it to PLOS ONE's style requirements.

2. Comment: Thank you for stating the following in the Competing Interests section: 'The authors have declared that no competing interests exist.' We note that one or more of the authors are employed by a commercial company: Independent research.

2.1. Please provide an amended Funding Statement declaring this commercial affiliation, as well as a statement regarding the Role of Funders in your study. If the funding organization did not play a role in the study design, data collection and analysis, decision to publish, or preparation of the manuscript and only provided financial support in the form of authors' salaries and/or research materials, please review your statements relating to the author contributions, and ensure you have specifically and accurately indicated the role(s) that these authors had in your study. You can update author roles in the Author Contributions section of the online submission form. Please also include the following statement within your amended Funding Statement. “The funder provided support in the form of salaries for authors [insert relevant initials], but did not have any additional role in the study design, data collection and analysis, decision to publish, or preparation of the manuscript. The specific roles of these authors are articulated in the ‘author contributions’ section.” If your commercial affiliation did play a role in your study, please state and explain this role within your updated Funding Statement.

Response and change: We thank the Editor’s comment. We have updated our Funding Statement in the Financial Disclosure on the submission system. 

Original text:

This work was financially supported by the Programa Nacional de Innovación Agraria (PNIA) of the Peruvian Government under the contract PI042-INIA-PNIA/UPMSI/IE.

 Revised text:

“This study was financially supported by Programa Nacional de Innovación Agraria (PNIA) of the Peruvian Government (http://www.pnia.gob.pe/) under the contract 042_PI. The funder provide support in the form of salaries for JF and HF and supplied materials for the study. Additionally, PM was supported by Independent Research in the form of salary. The funders had no role in study design, data collection and analysis, decision to publish, or preparation of the manuscript. The specific roles of these authors are articulated in the 'author contributions' section”. 

2.2. Please also provide an updated Competing Interests Statement declaring this commercial affiliation along with any other relevant declarations relating to employment, consultancy, patents, products in development, or marketed products, etc. Within your Competing Interests Statement, please confirm that this commercial affiliation does not alter your adherence to all PLOS ONE policies on sharing data and materials by including the following statement: "This does not alter our adherence to PLOS ONE policies on sharing data and materials.” (as detailed online in our guide for authors http://journals.plos.org/plosone/s/competing-interests). If this adherence statement is not accurate and there are restrictions on sharing of data and/or materials, please state these. Please note that we cannot proceed with consideration of your article until this information has been declared.

 Response and change: We thank the Editor’s comment. We have updated the Competing Interests Statement declaring our commercial affiliation (Competing Interest section on the submission system).

Original text:

The authors have declared that no competing interests exist.

Revised text: 

“PM is employed of Independent Research, and may own stock or hold stock options in that company. This does not alter the authors' adherence to PLOS ONE policies on sharing data and materials”.

2.3. Please include both an updated Funding Statement and Competing Interests Statement in your cover letter. We will change the online submission form on your behalf. Please know it is PLOS ONE policy for corresponding authors to declare, on behalf of all authors, all potential competing interests for the purposes of transparency. PLOS defines a competing interest as anything that interferes with, or could reasonably be perceived as interfering with, the full and objective presentation, peer review, editorial decision-making, or publication of research or non-research articles submitted to one of the journals. Competing interests can be financial or non-financial, professional, or personal. Competing interests can arise in relationship to an organization or another person. Please follow this link to our website for more details on competing interests: http://journals.plos.org/plosone/s/competing-interests

 Response and change: We thank the Editor’s comment. We have included both an updated Funding Statement and a Competing Interests Statement in the cover letter.

2.4. Please include captions for your Supporting Information files at the end of your manuscript, and update any in-text citations to match accordingly. Please see our Supporting Information guidelines for more information: http://journals.plos.org/plosone/s/supporting-information

Response and change: We agreed with the comment and we have included a list of supporting information and we updated the citations in the manuscript. 

i) Text added (page 21; lines 573-574)

Supporting information captions

S1 Database Morphological - yield traits and SSRs_Camacho-Villalobos A et al 2020.

S2 File.

ii) Update citations to supporting information: 

Material and Methods section; page 7; line 163: The complete collected phenotyping data is presented in S1 file. 

Material and Methods section; page 8; line 195: The complete raw genetic data related to SSRs is showed in S1 file.

Results section; page 10; lines 229-231: The PCA analysis demonstrated that both the morphological and yield traits explained the variability in the oil palm individuals in this study (S1 Table in S2 File).

Results section; page 11; lines 267-273: The frequency of null alleles was estimated between 0.01 and 0.48 for nine microsatellites (S2 Table in S2 File). Then, five microsatellites that presented null alleles were excluded to perform a genotype accumulation curve and expected heterozygosity tests to identify the effect of null alleles in the subsequent analysis. The analysis showed that the use of at least seven microsatellites could identify the totality of the 72 multilocus genotypes identified (S1 Fig in S2 File), also the inclusion or exclusion of the null alleles in the analysis did not alter the final He (p < 0.05) in the Peruvian germplasm (S2 Fig and S3 Table in S2 File).

Discussion section; page 15; lines 363-367: Null alleles were previously reported in Peruvian samples with frequencies around 0.06 and 0.21 (1), and in this study were reported between 0.01 to 0.48, however, the presence of five microsatellites with more than 10% of null alleles (or missing information) do not alter the estimation of the genetic diversity, in terms of He, of the germplasm of 72 samples. (S3 Table in S2 File). 

Discussion section; page 17; lines 420-422: … according to our analysis could be necessary at least 7 microsatellites in the Peruvian germplasm to carry up the complete analysis of genetic diversity (S1 Fig and S2 Table in S2 File), however many studies are necessary to corroborate these findings.

Response to Reviewer 1:

1. Comment 1: The study focuses on morphological, yield traits and genetic diversity of E.oleifera. However, the study on fruit related traits, which is an important commercial trait in oil palm is missing. Furthermore, according to descriptive statistics (Table 2), morphological traits has reduced variability than yield traits whereas yield traits has reduced variability than morphological traits in PCA analysis. There was a conflict between these two analyses. Is there any explanation for the observed results?

Response and change: We thank the reviewer for the comment. 

We have focused the main idea of this article in the principal morphological and yield traits and some yield traits were related with fruit such as: i) the weight of fruits at harvest, ii) the number of fruits per palm during harvest, iii) weight in kg of 10 fruits, and iv) fruit diameter (Material and Method section; Phenotyping). These traits were described in American oil palm for genetic studies using molecular markers such as SSRs, in order to set up future studies in Peru that figure out the relationship between morphological and yield traits in this important crop using molecular markers. A similar study was recently published in 2019 by Osorio-Guarín J. et al. using SNPs https://doi.org/10.1186/s12870-019-2153-8. However, we have added some lines to refer this limitation in our study in the initial version of this manuscript (Discussion section; page 13; lines 320-322. Additionally, we count on with the fruit related traits and a new manuscript is being prepared to present information related the fruit traits and quality in oil belong from this peruvian E.oleifera germplasm.

Regarding with the next sentence: “Furthermore, according to descriptive statistics (Table 2), morphological traits has reduced variability than yield traits whereas yield traits has reduced variability than morphological traits in PCA analysis. There was a conflict between these two analyses. Is there any explanation for the observed results?”. We have rephrased the whole paragraph in order to avoid the misinterpretation the results of variability. We emphasized in the results of descriptive statistics (Table 2) and PCA analysis, also we added a PCA results as a supplementary material to support our findings (S1 Table in S2 File).

Original text:

Descriptive statistics for the morphological and yield traits of the Peruvian Elaeis oleifera

germplasm (n=72) are shown in Table 2. The morphological trait that exhibited the highest standard deviation (sd) was FA (35.48%, 13 073.0 – 127 688.0 cm2) compared with TH and CC, wich had standard deviations of 11.65% and 11.89%, respectively. Moreover, four of the five yield traits showed high variability (22.10 – 46.91%). These data show that the Peruvian E.oleifera germplasm has reduced variability compared with that of its yield traits.

To compare the variability among traits, a principal component analysis (PCA) was performed to identify the main traits that contributed to the dispersion of the values (Fig 1). Of the total variation in the Peruvian E.oleifera germplasm, was explained by two components: PC1 (26.01%), which was associated with four morphological traits (FA, LEL, LXL and LD) and two yield traits (YP and ABW) and PC2, which was primarily associated with two morphological traits (TH and CC) and explained the 40.39% of the variance in the Peruvian germplasm. The PCA analysis demonstrated that the morphological traits explained more of the variability in the oil palm individuals in this study than the yield traits.

Revised sentence: Results section; page 9-10; lines 215-231

Descriptive statistics for the morphological and yield traits of the Peruvian Elaeis oleifera germplasm (n=72) are shown in Table 2. The morphological trait that exhibited the highest standard deviation (SD) was FA (SD: 20 894; min:13 073.0 – max: 127 688.0 cm2; cv: 35.48%) and, one (YP) of the five yield traits showed high standard deviation (SD: 22.76; min: 0.6 – max: 22.60 mt; cv: 46.91%). These data show that both morphological and yield traits presented variability in the Peruvian germplasm. 

A principal component analysis (PCA) was performed to identify the main traits that contributed to the dispersion of the values (Fig 1). Of the total variation in the Peruvian E.oleifera germplasm, was explained by two components: PC1 (26.01%), which was associated with five morphological traits (FA, LEL, LL, LXL, and LDW) and two yield traits (YP and ABW) and PC2, which was primarily associated with three morphological traits (TH, TD, and CC) and one yield trait (FW) explained the 40.39% of the variance in the Peruvian germplasm. The PCA analysis demonstrated that both the morphological and yield traits explained the variability in the oil palm individuals in this study (S1 Table in S2 File).

2. Comment 2. The author described the importance of relationship between the morphological traits and genetic diversity. According to their results, groups 1 and 3 were more closely related in morphological and yield traits whereas groups 1 and 2 were more closely related in genetic diversity analysis (number of alleles). Would appreciate more explanation for these results.

Response and change: We appreciate the revisor´s comment. Regarding the genetic differences between groups 1, 2 and 3 (number of alleles), we had tested that difference to He and A between the groups using statistic parameter and we had not found any evidence to the differentiation between groups and we reported it in the first version of this manuscript Discussion section; page 16; lines 390-394. However, we have considered to include some lines to explain the lack of differences due to the low number of individuals for each group. And, we would like to emphasize in the use of neutral microsatellites to identified a correct set of diversity genetic parameters. Discussion section; page 17; lines 403-405. 

3. Comment 3. The abstract can be made more reflective of the total paper. The authors need to highlight what are altogether the novel results of their study.

Response and change: We thank the reviewer for the comment. We proceed to improve the abstract in order to highlight our novel results and its impact in the Peruvian industry. Abstract section; pages 2-3; lines 46-71.

4. Comment 4. Although previously published methodologies were referred, the methods and the number of replicates for measurement of phenotypic data in materials and methods section should be briefly described.

Response and change: We thank the reviewer for the comment. we would like to rephrase a paragraph related to “Phenotyping” and add more information about the methods and the number of replicates that we employed in the study. 

Original text:

All experimental data obtained were related to the morphological and yield characteristics of an American palm (Elaeis oleifera H.B.K. Cortes) plantation that was laid out using a traditional pattern, a quincunx, with plant and row spacing of 9 x 9 m. Nine morphological characters were quantified: Trunk Diameter (TD, trunk circumference at the midsection), Trunk Height (HT, distance between the lowest green leaves and the fruit), Cup Coverage (CC, Leaf number), Leaf Length (LL, the rachis length of leaf number 17), Petiole Length (PL), Leaf Dry Weight (LDW, mean dry weight per leaf multiplied by the number of leaves produced), Foliar Area (FA, mean area per leaf multiplied by the number of leaves per palm), Leaflet length (LL, length of the largest leaflet), Leafleat diameter (LD) and Leaflet per Leaf (LXL, number of leaflet per leaf). The five yield characters were: Bunch Weight (ABW, the weight of fruits at harvest), Bunch Number (BN, the number of fruits per palm during harvest), the 10-weight fruits (FW, weight in kg of 10 fruits), fruit diameter (FD) and Yield per Palm (YP, kg of fruit per palm per year). Each measure was made using previously published methodologies (2, 20, 21).

Revised sentence: Material and Methods section; page 6-7; lines 134-163

All experimental data obtained were related to the morphological and yield characteristics of an American palm (Elaeis oleifera H.B.K. Cortes) plantation that was laid out using a traditional pattern, a quincunx, with plant and row spacing of 9 x 9 m. Nine morphological characters were quantified: Trunk Diameter (TD was taken from the trunk circumference at the midsection with a tape from 10 cm below the crown height), Trunk Height (HT was estimated from the distance between the lowest green leaves and the fruit), Cup Coverage ((CC) The leaves were counted one by one following the phyllotaxis of the trunk), Leaf Length ((LL) the leaf located in the middle third was taken, similar to the position of leaf number 17 in E. guineensis, once it was cut, the total length of petiole and length of the rachis were added to estimate the Leaf Length)), Petiole Length (PL was measured from petiole separated of the rachis in leaf number 17), Leaf Dry Weight (The dry weight were determined by chopping the leaf number 17 finely, and drying to constant temperature at 100-105ºC, them, the dry weight was multiplied by the number of leaves produced to estimate LDW), Foliar Area (FA represents the mean area per leaf multiplied by the number of leaves per palm this measure was obtained by measuring two leaves per palm), Leaflet length (LL was determinates in the largest leaflet in the leaf number 17 by measuring from apex to the insertion base of the rachis leaf ), Leaflet diameter (LD was estimated by bending the largest leaflet in the middle and then width measure was taken) and Leaflet per Leaf (LXL, the number of leaflet per leaf was estimated by counting the leaflets in the leaf number 17 including rudimentary leaflets). The five yield characters were: Average Bunch Weight (ABW was determined by weighing the bunch at harvest time) , Bunch Number (BN was represented by the number of bunch per palm collected during the first harvest), the 10-weight fruits (FW was estimated by weighing 10 fruits at random), fruit diameter (FD) and Yield per Palm (YP, this indicator was calculated form the total production in kg of fruit per palm per year). Each measure was made using previously published methodologies (2, 20, 21). And, the number of replications for each morphological trait were done every two weeks during the year of evaluation with the exception of the LL, PL, LDW, and LD traits that were done once due to the implication of the use of destructive steps. The complete collected phenotyping data is presented in S1 file.

5. Comment 5. In the results section (Page 8, Line #186-187), it is not clear whether yield traits are compared to morphological traits or not. Please re-write that part.

Response and change: We thank the reviewer for her/his observation. We agree with the reviewer; however, we decided eliminate this sentence in order to make a response to Reviewer 1, comment 1. 

6. Comment 6. Page 11, Line # 261, the word "wich" should be "which"..).

Response and change: We thank the reviewer for the correction. We agree and we proceed to corrected it. This change is shown in yellow color. Discussion section; page 13; line 307

7. Comment 7. Page 581, Table 4, the word "morphologycal" should be "morphological".

Response and change: We thank the reviewer for the correction. We agree and we proceed to corrected it. This change is shown in yellow color. Table 4

8. Comment 8. Page 262, Line 262-263, please clearly describe the sentences.

Response and change: We thank the reviewer for the observation. Also, we considered to add a sentence to explain the differences between Colombian and Peruvian germplasm.

Original text:

The primary phenotypic differences within this Peruvian germplasm were based on two morphological traits, trunk height (TH) and cup coverage (CC), wich explained 40.4% of the total variation in contrast, the variation in an Elaeis oleifera Colombian germplasm was explained by for four traits (trunk height, leaf area, foliar area and leaf dry weight) (2)

Revised sentence: Discussion section; page 13; lines 305-310

The primary phenotypic differences within this Peruvian germplasm were based on three morphological traits, trunk height (TH), trunk diameter (TD) and cup coverage (CC), and one yield trait (FW, Fruits 10-weight)which explained 40.4% of the total variation in contrast, the variation in an E. oleifera Colombian germplasm was explained by four morphological traits (trunk height, leaf area, foliar area, and leaf dry weight) (2) however, the age of both plantations could explain this differences.

9. Comment 9. English is sometimes odd, please improve it.

Response and change: We thank the reviewer for his/her comment. We considered improving the English employing a professional language service. We attached the certificate of this service.

Response to Reviewer 2:

1. Comment 1. The manuscript describes genetic diversity of Elaeis oleifera assembled from an area in Peru by means of 12 SSRs and selected morphological traits. The manuscript was written in readable English language but with minor spelling mistakes and some grammatical errors. These errors are however correctable. I have specific concerns as below. The phenotypic data was collected at very young age, when the palms are 4 years old, as described by authors in method section. It should be cautioned that analysis and interpretation of such data can be erroneous and misleading. It is very important to note that For oil palm specifically, standard phenotypic data collection is carried out for at least 8 years, starting from palms at 2 or 3 years old. Measuring vegetative traits such as height, frond length is carried out once on palms at 8th year old. I am doubtful on the results presented especially if it is going to be used for the purposes of selection and breeding.

Response and change: We thank the reviewer for his/her comment. Regarding to the English language we answered on the same way that the reviewer 1, comment 9. 

Regarding with the next sentence: “The phenotypic data was collected at very young age, when the palms are 4 years old, as described by authors in method section. It should be cautioned that analysis and interpretation of such data can be erroneous and misleading. It is very important to note that For oil palm specifically, standard phenotypic data collection is carried out for at least 8 years, starting from palms at 2 or 3 years old. Measuring vegetative traits such as height, frond length is carried out once on palms at 8th year old. I am doubtful on the results presented especially if it is going to be used for the purposes of selection and breeding”. 

We agree with the reviewer and we have done some changes in the message of our manuscript in order to provide a realistic application of our results. 

i) We clarified that the term “four-year-old” of the germplasms is since of transplanting event. Material and methods section; page 6; lines 129-130. 

ii) The objective of our study was rephrased as follow:

Original Text

In that context, the aim of this study was to characterize the E. oleifera (H.B.K) Cortes germplasm of a Peruvian collection using morphological traits and SSR markers to further national breeding programs for the oil palm.

Revised sentence: Introduction section; page 5; lines 118-121

In that context, the aim of this study was to characterize the E. oleifera (H.B.K) Cortes germplasm of a Peruvian collection using morphological traits and SSR markers to contributed with important data for setting up of further studies that will be related to the development of national breeding programs for the oil palm.

iii) We eliminated a sentence and added a new paragraph to highlight the principal limitation of our study related to the age of the germplasm.

Eliminated Text: Discussion section; page 17; lines 429-431

This information must guide us in the decision-making process when planning breeding programs that are focused on crosses to obtain OxG interspecific hybrids (E. oleifera x E. guineensis) with interesting yield traits.

Added text: Discussion section; page 18; lines 435-438

However, this information should be taken carefully due the age of the germplasm (four-year-old) and further studies are necessary to complement it, by including new data trait (morphological and yield) corresponding to the next four years in according to other previous studies (1,20).

iv) The final paragraph of our study was rephrased as follow:

Original Text

Finally, the information presented here demonstrates that the 72 accessions of the Peruvian E. oleifera located in Ucayali region of Peru represent a useful genetic resource to promote the use of the genetic material and to design new national breeding strategies and conservation programs for this important crop in the tropical regions of Peru and South America.

Revised sentence: Discussion section; page 18; lines 431-435

Finally, this novel information presented here demonstrates that the 72 accessions of the Peruvian E. oleifera located in Ucayali region of Peru represent a useful genetic resource to promote the use of the genetic material and to promote and design further investigations to establish national breeding strategies and conservation programs for this important crop in the tropical regions of Peru and South America.

v)Additional revisions were made by the authors in the discussion section to make clear the scope of our results. Discussion section; page 12; lines 293-295 and pages 15-16; lines 375-377.

2. Comment 2. The challenge of using SSRs for genetic analysis is the occurrence of null alleles. Null alleles introduce error in assessment of genetic diversity, parentage analysis etc. The authors can carry out relevant analysis to identify null alleles, exclude them from dataset prior to genetic diversity analysis.

Response and change: We thank the reviewer for his/her comment. We agree with the comment and we tested the impact of null alleles in our analysis. Our additional analysis is described in the material and methods section and the supplementary data section, we found that five SSRs have null alleles and we excluded them and performed comparative tests between all complete set (12 SSRs) and 7 SSRs (without 5 SSRs), and our results showed that there are no significant differences. At the light of these results we decided to include this new information throughout the manuscript as follows: 

i) We added a new paragraph in Material and methods section (Materials and methods section; page 9; lines 207-211).

Additionally, to test the effect of null alleles on the power of discrimination genotypes was addressed by the construction of a genotype accumulation curve with 1,000 iterations with replacement of samples, while the effect on the expected heterozygosity was addressed by a Monte Carlo test with 1,000 loci resampling considering the exclusion of loci with more than 10% of null alleles. All these procedures were tested using the R v3.42 software.

ii) We added a new paragraph in Results section (Results section; page 11; lines 267-273).

The frequency of null alleles was estimated between 0.01 and 0.48 for nine microsatellites (S1 Table in S2 File). Then, five microsatellites that presented null alleles were excluded to perform a genotype accumulation curve and expected heterozygosity tests to identify the effect of null alleles in the subsequent analysis. The analysis showed that the use of at least seven microsatellites could identify the totality of the 72 multilocus genotypes identified (S1 Fig in S2 File), also the inclusion or exclusion of the null alleles in the analysis not alter the final He (p < 0.05) in the Peruvian germplasm (S2 Fig and S3 Table in S2 File).

iii) We added a new paragraph in Discussion section (Discussion section; page 15; lines 363-370).

Null alleles were previously reported in Peruvian samples with frequencies around 0.06 and 0.21 (1), and in this study were reported between 0.01 to 0.48, however, the presence of five microsatellites with more than 10% of null alleles (or missing information) do not alter the estimation of the genetic diversity, in terms of He, of the germplasm of 72 samples. (S3 Table in S2 File). These alleles are produced by putative mutations that occur at the microsatellite binding sites that could prevent primer annealing and in consequence, the lack of amplification of the PCR products (8) therefore further primer development could be necessary for the local or regional E. oleifera germplasm.

iv) We considered to add a Supporting Information file to support our additional results related to null alleles (S2 table, S3 table, S1 Fig and S2 Fig) in according to reviewer 2, comment 2.

3. Comment 3. The number of SSRs applied is limited. The trend of publication nowadays applied hundreds or even thousands of markers for diversity analysis. I encourages authors to add more markers to achieve reasoanably acceptable statistics to support any conclusions made from the analysis.

Response and change: We really appreciate the reviewer for his/her recommendation. In order to achieve an evidence about the utility of our data, we performed an analysis by the construction of a genotype accumulation curve with 1,000 iterations with replacement of samples employed (n=72). Our results show us that the use at least 7 SSRs could show us the correct identification of 72 multilocus genotypes. Also, we identified two recent studies carried out in central (Ithnin M et al 2017 DOI 10.1186/s12863-017-0505-7) and south America (Arias D et al 2015 DOI 10.1007/s11295-015-0946-y) where were employed a small SSRs set (<20). In our study we employed twelve microsatellites and we could identify the 72 multilocus genotypes in the Peruvian E. oleifera germplasm according to our additional and exhaustive analysis. 

Additionally, the large set of SSRs is necessary to the identification of QTLs associated with any special trait in oil palm and segregation studies for example Jeennor S and Volkaert S 2014 (DOI 10.1007/s11295-013-0655-3) reported the use of 83 SSR and 101 SNPs, and Osorio-Guarín J et al 2019 (doi.org/10.1186/s12870-019-2153-8) reported the use of 3776 SNPs without SSRs to identified QTLs in E. oleifera. For that in special to E.oleifera diversity genetic the use of SSRs at least in South America will tend to be constant, however, the use of new markers such as SNPs will be restricted to QTL identification according to recent reports (Osorio-Guarín J et al 2019). 

At the light of these results we decided to include this new information throughout the manuscript as follows:

i) We added a new paragraph in Results section (Results section; page 11; lines 267-273).

The frequency of null alleles was estimated between 0.01 and 0.48 for nine microsatellites (S2 Table in S2 File). Then, five microsatellites that presented null alleles were excluded to perform a genotype accumulation curve and expected heterozygosity tests to identify the effect of null alleles in the subsequent analysis. The analysis showed that the use of at least seven microsatellites could identify the totality of the 72 multilocus genotypes identified (S1 Fig in S2 File), also the inclusion or exclusion of the null alleles in the analysis not alter the final He (p < 0.05) in the Peruvian germplasm (S2 Fig and S3 Table in S2 File).

ii) We added lines to a paragraph in Discussion section (Discussion section; page 17; lines 417-422).

Moreover, the limited economical resources in many laboratories in South America could interrupt the adoption of these molecular tools. Recently, there are studies in E.oleifera that show that useful information is achieved when is employed a limited set of microsatellites (<20 SSRs) (1,8) and according to our analysis could be necessary at least 7 microsatellites in the Peruvian germplasm to carry up the complete analysis of genetic diversity (S1 Fig and S3 Table in S2 File), however many studies are necessary to corroborate these findings.

iii) We considered to add a Supporting Information file (S1 Fig. in S2 file) to support our additional results related to null alleles in according to reviewer 2, comment 3.

iv) We would considerate to share the all database in order to allow the discussion between distinct groups about the data generated in this study. We added S1 file in supporting information section.

---

## [Decision Letter · Decision Letter 1]

3 Sep 2020

PONE-D-20-17719R1

Morphological and molecular characterization of an Elaeis oleifera (H.B.K) cortes germplasm collection located in Ucayali, Peru.

PLOS ONE

Dear Dr. Bendezu,

Thank you for submitting your manuscript to PLOS ONE. After careful consideration, we feel that it has merit but does not fully meet PLOS ONE’s publication criteria as it currently stands. Therefore, we invite you to submit a revised version of the manuscript that addresses the points raised during the review process.

We look forward to receiving your revised manuscript.

Kind regards,

Tzen-Yuh Chiang

Academic Editor

PLOS ONE

Reviewers' comments:

Reviewer's Responses to Questions

**Comments to the Author**

1. If the authors have adequately addressed your comments raised in a previous round of review and you feel that this manuscript is now acceptable for publication, you may indicate that here to bypass the “Comments to the Author” section, enter your conflict of interest statement in the “Confidential to Editor” section, and submit your "Accept" recommendation.

Reviewer #1: All comments have been addressed

Reviewer #2: (No Response)

2. Is the manuscript technically sound, and do the data support the conclusions?

Reviewer #1: Yes

Reviewer #2: No

3. Has the statistical analysis been performed appropriately and rigorously? 

Reviewer #1: Yes

Reviewer #2: Yes

4. Have the authors made all data underlying the findings in their manuscript fully available?

Reviewer #1: Yes

Reviewer #2: Yes

5. Is the manuscript presented in an intelligible fashion and written in standard English?

Reviewer #1: Yes

Reviewer #2: No

6. Review Comments to the Author

Reviewer #1: The revised version of the research article has been improved based on the comments and suggestions from the reviewers. However, please recheck the English of the manuscript. There are still some minor grammatical adjustments to be made.

Reviewer #2: (No Response)

7. PLOS authors have the option to publish the peer review history of their article (what does this mean?). If published, this will include your full peer review and any attached files.

Reviewer #1: No

Reviewer #2: No

---

## [Author Response · Author response to Decision Letter 1]

28 Sep 2020

6. Review Comments to the Author

Response to Reviewer 1:

1. Comment 1: The revised version of the research article has been improved based on the

comments and suggestions from the reviewers. However, please recheck the English of the

manuscript. There are still some minor grammatical adjustments to be made.

Response and change: We thank the reviewer for the comment. We have checked and

corrected the minor grammatical adjustments in the manuscript. Also, we completed filiation

information in the author’s section (lines 7-17).

---

## [Decision Letter · Decision Letter 2]

29 Oct 2020

PONE-D-20-17719R2

Morphological and molecular characterization of an Elaeis oleifera (H.B.K) cortes germplasm collection located in Ucayali, Peru.

PLOS ONE

Dear Dr. Bendezu,

Thank you for submitting your manuscript to PLOS ONE. After careful consideration, we feel that it has merit but does not fully meet PLOS ONE’s publication criteria as it currently stands. Therefore, we invite you to submit a revised version of the manuscript that addresses the points raised during the review process.

We look forward to receiving your revised manuscript.

Kind regards,

Tzen-Yuh Chiang

Academic Editor

PLOS ONE

Reviewers' comments:

Reviewer's Responses to Questions

**Comments to the Author**

1. If the authors have adequately addressed your comments raised in a previous round of review and you feel that this manuscript is now acceptable for publication, you may indicate that here to bypass the “Comments to the Author” section, enter your conflict of interest statement in the “Confidential to Editor” section, and submit your "Accept" recommendation.

Reviewer #1: All comments have been addressed

Reviewer #2: (No Response)

Reviewer #3: (No Response)

2. Is the manuscript technically sound, and do the data support the conclusions?

Reviewer #1: Yes

Reviewer #2: No

Reviewer #3: Yes

3. Has the statistical analysis been performed appropriately and rigorously? 

Reviewer #1: Yes

Reviewer #2: No

Reviewer #3: Yes

4. Have the authors made all data underlying the findings in their manuscript fully available?

Reviewer #1: Yes

Reviewer #2: Yes

Reviewer #3: Yes

5. Is the manuscript presented in an intelligible fashion and written in standard English?

Reviewer #1: Yes

Reviewer #2: No

Reviewer #3: Yes

6. Review Comments to the Author

Reviewer #1: The manuscript has been clearly improved as suggested by the reviewers and can be accepted for publication.

Reviewer #2: As I have indicated in my previous reports, I have two major concerns on the manuscript:

1. The number of markers used (72 SSRs) are too little and does not provide a reliable outcome of analysis. I suggest that the authors add more markers as the current technologies allows hundreds and thousands of markers to be genotyped across thousands of samples in a single experiment. It is important to keep pace with the current trend of publications where results from thousands of marker are presented and discussed.

2. The phenotypic data are collected at young age of palms (4 year old) and this does not fulfill the standard data collection of palms ie. 8th year for vegetative traits (eg. trunk diameter, trunk height, leaf length etc.) and at least 4 consecutive year for yield parameters (eg. bunch weight, bunch number). The authors would require at least another 4/5 years to collect such data and add to the manuscript.

I hope the authors address my comments as above.

Reviewer #3: This represents a comprehensive study through morphological and molecular marker-based genetic diversity characteristaion useful for future breeding program and establishment of core germplasm collection of the E. oleifera oil palm species of Peruvian origin. Discussion include comparison with previous studies using materials from other South American regions on traits of importance contributing to variations and information revealed by the specific microsatellite markers. Phenotypic groups were identified as potential collection for future genetic improvement.

7. PLOS authors have the option to publish the peer review history of their article (what does this mean?). If published, this will include your full peer review and any attached files.

Reviewer #1: No

Reviewer #2: No

Reviewer #3: No

---

## [Author Response · Author response to Decision Letter 2]

10 Dec 2020

Response to Reviewer 1:

1. Comment 1: The manuscript has been clearly improved as suggested by the reviewers and can be accepted for publication.

Response: We thank the reviewer for the comment.

 Response to Reviewer 2:

1. Comment 1: As I have indicated in my previous reports, I have two major concerns on the manuscript: The number of markers used (72 SSRs) are too little and does not provide a reliable outcome of analysis. I suggest that the authors add more markers as the current technologies allows hundreds and thousands of markers to be genotyped across thousands of samples in a single experiment. It is important to keep pace with the current trend of publications where results from thousands of marker are presented and discussed.

Response and change: We thank the reviewer for the comment. However, that comment was answered for the authors in the first rebuttal letter in August 2020 and we highlighted this idea in the manuscript (Discussion section, lines 385-393). Actually, we have revised references that pointed up the use of SSRs in numbers around to 20 markers applied on E. Oleifera germplasm from Central and South America (Maizura Ithnin et al. BMC Genetics (2017) 18:37 DOI 10.1186/s12863-017-0505-7 and Diana Arias et al. Tree Genetics & Genomes (2015) 11:122 DOI 10.1007/s11295-015-0946-y) and we had performed additional analysis where we demonstrated that could be necessary at least 7 microsatellites in the Peruvian germplasm to carry up the complete analysis of genetic diversity (File S2). We know about the new technologies such as SNPs that allow evaluating hundreds of markers applied in our region to address Genome-wide association study (Osorio-Guarín et al. BMC Plant Biology (2019) 19:533 https://doi.org/10.1186/s12870-019-2153-8), however with a limited resource to performed that technologies, we looked up in the previous state of the art and identified the Maizura Ithnin and Diana Arias reports applied in South American and Peruvian resources. Additionally, in our first rebuttal letter in August 2020, we had considered to add a sentence to perform further studies employing SNPs technologies following the Colombian experience (Discussion section, lines 381-384). 

We would like to emphasize that the main objective of our manuscript is to contribute with preliminary data in Peru to perform further assays in E.oleiferera germplasm using technologies applied in our region and we revised that idea in our objective and conclusion (Introduction section, line 100; Discussion section, line 408). 

2. Comment 2: The phenotypic data are collected at young age of palms (4 year old) and this does not fulfill the standard data collection of palms ie. 8th year for vegetative traits (eg. trunk diameter, trunk height, leaf length etc.) and at least 4 consecutive year for yield parameters (eg. bunch weight, bunch number). The authors would require at least another 4/5 years to collect such data and add to the manuscript.

Response: We really appreciate the reviewer for his/her recommendation. Taking into account the recommendation done in the first revision (first rebuttal letter in August 2020), we had added mainly the limitation of the use of our preliminary data due the age of the palms in our last version (Discussion section, lines 400-406). We considered the importance of this preliminary novel finding to Peru to set up further studies in E.oleifera and it will have an impact on the local oil palm industry. Finally, we will complement these findings with novel raw data that we have collected since 2019 to preparade new manuscripts about the Peruvian germplasm.

Response to Reviewer 3:

1.- Comment 1: This represents a comprehensive study through morphological and molecular marker-based genetic diversity characterization useful for future breeding program and establishment of core germplasm collection of the E. oleifera oil palm species of Peruvian origin. Discussion include comparison with previous studies using materials from other South American regions on traits of importance contributing to variations and information revealed by the specific microsatellite markers. Phenotypic groups were identified as potential collection for future genetic improvement.

Response: We thank the reviewer for the comment.

---

## [Decision Letter · Decision Letter 3]

17 Dec 2020

PONE-D-20-17719R3

Morphological and molecular characterization of an Elaeis oleifera (H.B.K) cortes germplasm collection located in Ucayali, Peru.

PLOS ONE

Dear Dr. Bendezu,

Thank you for submitting your manuscript to PLOS ONE. After careful consideration, we feel that it has merit but does not fully meet PLOS ONE’s publication criteria as it currently stands. Therefore, we invite you to submit a revised version of the manuscript that addresses the points raised during the review process.

We look forward to receiving your revised manuscript.

Kind regards,

Tzen-Yuh Chiang

Academic Editor

PLOS ONE

Reviewers' comments:

Reviewer's Responses to Questions

**Comments to the Author**

1. If the authors have adequately addressed your comments raised in a previous round of review and you feel that this manuscript is now acceptable for publication, you may indicate that here to bypass the “Comments to the Author” section, enter your conflict of interest statement in the “Confidential to Editor” section, and submit your "Accept" recommendation.

Reviewer #2: (No Response)

Reviewer #3: All comments have been addressed

2. Is the manuscript technically sound, and do the data support the conclusions?

Reviewer #2: No

Reviewer #3: Yes

3. Has the statistical analysis been performed appropriately and rigorously? 

Reviewer #2: Yes

Reviewer #3: Yes

4. Have the authors made all data underlying the findings in their manuscript fully available?

Reviewer #2: Yes

Reviewer #3: Yes

5. Is the manuscript presented in an intelligible fashion and written in standard English?

Reviewer #2: Yes

Reviewer #3: Yes

6. Review Comments to the Author

Reviewer #2: Dear Authors,

For the third time, I highlight the two important elements that need to be improved in the manuscript

1. The number of markers used need to be increased. My recommendation is based on papers published in PlosOne for example in https://doi.org/10.1371/journal.pone.0224763, where around 1.5K SNPs were applied for diversity study. I believed this is the number expected in PlosOne. I understand the convincing attempt made by the authors by quoting similar papers published in BMC Genetics and Tree Genetics and Genome. However, these journals have their own standards, which is different from PlosOne.

2. My second concern is on the data collection which is one of the important elements that reflects the quality of scientific papers. As I have indicated twice before, at least 4 consecutive year of data collection is needed. Even if these are satisfied, the data consist of early fruit bearing years and therefore is not conclusive for selection and breeding purposes.

I have clearly indicated my concerns, as above. I urge the authors to improve the manuscript as recommended, before the manuscript can be submitted for review.

Reviewer #3: Can be accepted for publication. This is a significant contribution to the work on the source of breeding materials for oil palm improvement. The work on Elaeis oleifera should be emphasized in South America as the African oil palm, Elaeis guineensis is not suitable for cultivation in this region. The authors appeared to have addressed all the concerns posted by the reviewers. I have no objection on acceptance since I have agreed to accept the R2 version.

7. PLOS authors have the option to publish the peer review history of their article (what does this mean?). If published, this will include your full peer review and any attached files.

Reviewer #2: No

Reviewer #3: No

---

## [Author Response · Author response to Decision Letter 3]

1 Mar 2021

Response to Reviewer 2:

1. Comment 1: The number of markers used need to be increased. My recommendation is based on papers published in PlosOne for example in https://doi.org/10.1371/journal.pone.0224763, where around 1.5K SNPs were applied for diversity study. I believed this is the number expected in PlosOne. I understand the convincing attempt made by the authors by quoting similar papers published in BMC Genetics and Tree Genetics and Genome. However, these journals have their own standards, which is different from PlosOne.

Response and change: First of all, we would like to thank the Reviewer #2 for the comment. We would like to mention that the link shared by the Reviewer corresponds to a scientific publication focused on cassava species, not on E. oleifera crop. Regarding this, we have addressed all of these questions in the rebuttal letters sent to the journal previously, we believe our answers were presented in a consistent and robust way. Likewise, the inclusion of SNP technology is too expensive for our institution in Peru at this moment. Unfortunately, the economic impact due to the COVID-19 pandemics has affected the financial resources of our institution narrowing our capacity to perform the assays requested by the Reviewer #2. Additionally, the publication of our results is part of a research project financially supported by the Peruvian Government; therefore, we believe that the preliminary data show in this manuscript will, indeed, encourage the development of further studies including the SNP analyses. 

Comment 2: My second concern is on the data collection which is one of the important elements that reflects the quality of scientific papers. As I have indicated twice before, at least 4 consecutive year of data collection is needed. Even if these are satisfied, the data consist of early fruit bearing years and therefore is not conclusive for selection and breeding purposes. I have clearly indicated my concerns, as above. I urge the authors to improve the manuscript as recommended, before the manuscript can be submitted for review.

Response: We really appreciate the reviewer for his/her recommendation. First of all, we would like to thank the Reviewer #2 for the comment. We would like to emphasize that the preliminary data showed in this manuscript will contribute and promote the development and performance of further assays in E. oleifera germplasm using technologies including SNP analyses. On the other hand, we have included a whole paragraph about the limitations of the study. In this paragraph, we addressed the Reviewer #2´s comments regarding the data collection time of germplasm. We would like to highlight that this is the first manuscript that shows preliminary data collected from E. oleifera (H.B.K) germplasm from the Peruvian jungle; therefore, this preliminary datawill set up further studies focused on E. oleifera in Peru, having an impact on the local oil palm industry. 

Response to Reviewer 3:

1.- Comment 1: Can be accepted for publication. This is a significant contribution to the work on the source of breeding materials for oil palm improvement. The work on Elaeis oleifera should be emphasized in South America as the African oil palm, Elaeis guineensis is not suitable for cultivation in this region. The authors appeared to have addressed all the concerns posted by the reviewers. I have no objection on acceptance since I have agreed to accept the R2 version.

Response: We thank the reviewer for the comment.

---

## [Decision Letter · Decision Letter 4]

19 Mar 2021

PONE-D-20-17719R4

Morphological and molecular characterization of an Elaeis oleifera (H.B.K) cortes germplasm collection located in Ucayali, Peru.

PLOS ONE

Dear Dr. Bendezu,

Thank you for submitting your manuscript to PLOS ONE. After careful consideration, we feel that it has merit but does not fully meet PLOS ONE’s publication criteria as it currently stands. Therefore, we invite you to submit a revised version of the manuscript that addresses the points raised during the review process.

We look forward to receiving your revised manuscript.

Kind regards,

Tzen-Yuh Chiang

Academic Editor

PLOS ONE

Journal Requirements:

Reviewers' comments:

Reviewer's Responses to Questions

**Comments to the Author**

1. If the authors have adequately addressed your comments raised in a previous round of review and you feel that this manuscript is now acceptable for publication, you may indicate that here to bypass the “Comments to the Author” section, enter your conflict of interest statement in the “Confidential to Editor” section, and submit your "Accept" recommendation.

Reviewer #4: (No Response)

2. Is the manuscript technically sound, and do the data support the conclusions?

Reviewer #4: Yes

3. Has the statistical analysis been performed appropriately and rigorously? 

Reviewer #4: Yes

4. Have the authors made all data underlying the findings in their manuscript fully available?

Reviewer #4: Yes

5. Is the manuscript presented in an intelligible fashion and written in standard English?

Reviewer #4: Yes

6. Review Comments to the Author

Reviewer #4: Dear authors and editor,

After 4 revisions, the manuscript is mature and almost ready for publication. Congratulations for the authors.

It could be better with oil quality analysis (demands different expertise and infrastructure) and/or with a more representative set of molecular markers, such as SNPs or array, but I fully understand and agree with the authors responses - it´s a huge effort just to keep the germplasm bank in good (not ideal) conditions also at south america, and in that context the presented work is relevant and well done. Elaeis guineensis and E. oleifera present a really low traditional breeding cycle, and usually far from ideal germplasm conditions, such as inconstant founding and low budgets. Consequently, a lot of variability can be lost over the years. Each step, even if not as complete and comprehensive as the state-of-the-art technology would allow, is important for the preservation of the variability of the species and for the improvement of the breeding programs.

Only one point remains to be corrected or explained for acceptance. At PCA results (both at text and figure, starting at page 8), the PC2 is bigger than PC1. I believe that the pattern is that PC1 explains most of the variance, PC2 the second largest and so on. Considering this pattern, I am also not sure if what was considered to be PC2 is in fact the value of the bigger component or the sum of the first two (PC1 + PC2).

Thanks you.

7. PLOS authors have the option to publish the peer review history of their article (what does this mean?). If published, this will include your full peer review and any attached files.

Reviewer #4: No

---

## [Author Response · Author response to Decision Letter 4]

28 Mar 2021

Journal Requirements:

Author Response: We checked and updated some links in the original reference list. Reference list section Lines 448, 455, 473, 489 and 514.

Response to Reviewer 4:

1. Comment 1: After 4 revisions, the manuscript is mature and almost ready for publication. Congratulations for the authors. It could be better with oil quality analysis (demands different expertise and infrastructure) and/or with a more representative set of molecular markers, such as SNPs or array, but I fully understand and agree with the authors responses - it´s a huge effort just to keep the germplasm bank in good (not ideal) conditions also at south America, and in that context the presented work is relevant and well done. Elaeis guineensis and E. oleifera present a really low traditional breeding cycle, and usually far from ideal germplasm conditions, such as inconstant founding and low budgets. Consequently, a lot of variability can be lost over the years. Each step, even if not as complete and comprehensive as the state-of-the-art technology would allow, is important for the preservation of the variability of the species and for the improvement of the breeding programs. 

Only one point remains to be corrected or explained for acceptance. At PCA results (both at text and figure, starting at page 8), the PC2 is bigger than PC1. I believe that the pattern is that PC1 explains most of the variance, PC2 the second largest and so on. Considering this pattern, I am also not sure if what was considered to be PC2 is in fact the value of the bigger component or the sum of the first two (PC1 + PC2).

Response: First of all, we would like to thank the Reviewer #4 for the comment. Regarding PCA concerns, we clarified that the PC2 value is the cumulative value from both PC1 and PC2, and we corrected it in the manuscript in order to avoid some misinterpretation (Results section, line 203, page 9), also we corrected the figure 1 and its legend (Line 552),

---

## [Editor Report · Decision Letter 5]

7 Apr 2021

Morphological and molecular characterization of an Elaeis oleifera (H.B.K) cortes germplasm collection located in Ucayali, Peru.

PONE-D-20-17719R5

Dear Dr. Bendezu,

We’re pleased to inform you that your manuscript has been judged scientifically suitable for publication and will be formally accepted for publication once it meets all outstanding technical requirements.

Kind regards,

Tzen-Yuh Chiang

Academic Editor

PLOS ONE
---

## [Editor Report · Acceptance letter]

16 Apr 2021

PONE-D-20-17719R5 

Morphological and molecular characterization of an *Elaeis oleifera* (H.B.K) Cortes germplasm collection located in Ucayali, Peru. 

Dear Dr. Bendezu:

I'm pleased to inform you that your manuscript has been deemed suitable for publication in PLOS ONE. Congratulations! Your manuscript is now with our production department. 

Kind regards, 

on behalf of

Dr. Tzen-Yuh Chiang 

Academic Editor

PLOS ONE